
# Mapping current and activity fluctuations in exclusion processes: consequences and open questions

Matthieu Vanicat[1,3], Eric Bertin[2], Vivien Lecomte[2] and Eric Ragoucy[1]

**1** LAPTh, CNRS, USMB, F-74000 Annecy, France
**2** Université Grenoble Alpes, CNRS, LIPhy, F-38000 Grenoble, France
**3** OSE Engineering, 66000 Perpignan, France

## Abstract

Considering the large deviations of activity and current in the Asymmetric Simple Exclusion Process (ASEP), we show that there exists a non-trivial correspondence between the joint scaled cumulant generating functions of activity and current of two ASEPs with different parameters. This mapping is obtained by applying a similarity transform on the deformed Markov matrix of the source model in order to obtain the deformed Markov matrix of the target model. We first derive this correspondence for periodic boundary conditions, and show in the diffusive scaling limit (corresponding to the Weakly Asymmetric Simple Exclusion Processes, or WASEP) how the mapping is expressed in the language of Macroscopic Fluctuation Theory (MFT). As an interesting specific case, we map the large deviations of current in the ASEP to the large deviations of activity in the SSEP, thereby uncovering a regime of Kardar–Parisi–Zhang in the distribution of activity in the SSEP. At large activity, particle configurations exhibit hyperuniformity [Jack *et al.*, PRL 114 060601 (2015)]. Using results from quantum spin chain theory, we characterize the hyperuniform regime by evaluating the small wavenumber asymptotic behavior of the structure factor at half-filling. Conversely, we formulate from the MFT results a conjecture for a correlation function in spin chains at any fixed total magnetization (in the thermodynamic limit). In addition, we generalize the mapping to the case of two open ASEPs with boundary reservoirs, and we apply it in the WASEP limit in the MFT formalism. This mapping also allows us to find a symmetry-breaking dynamical phase transition (DPT) in the WASEP conditioned by activity, from the prior knowledge of a DPT in the WASEP conditioned by the current.


# 1  Introduction

Fluctuations in random processes are well described by the theory of large deviations, whenever a large-size and/or a large-time asymptotics naturally comes into play (see e.g. [1] for a review). Recently the statistics of time-averaged observables (such as the particle current or the energy flow) has attracted much attention; the large-deviations settings allows one to characterize the probability distribution of such observables, for quantum [2–11] and classical systems [12–41]. In contrast to equilibrium statistical mechanics (where observables of interest depend only on the configuration of the system at a given time), the time-averaged observables depend on the full history of the system on a given time window. Yet, as in equilibrium, the distribution of such observables can present an abrupt change, characterized by a singularity in the large deviation function (LDF) that describes the large-time scaling of the distribution. Such dynamical phase transitions (DPTs) are present in a variety of problems, from driven diffusive systems [42–58], to models of glass formers [39,40,59–63], interfaces [64–68], and active matter [69–71]. In this paper, we study a model of particle transport on a lattice, uncovering a new scaling regime close to a DPT that is present in the model.

Exclusion processes constitute a paradigmatic class of particle transport models with minimalist interactions: particles jump on a lattice with rates possibly depending on the orientation of the jump, provided the target site of the jump is unoccupied. In spite of their simplicity, such models have provided a successful playground for the determination of non-equilibrium steady-states or current large deviations (see [12] for a review), and this in a non-perturbative way, far from equilibrium. A key aspect of their study is that their Markov operator of evolution

(which describes the linear evolution in time of the probability distribution) can be expressed as a spin chain operator [72–74]. The LDFs of interest can be computed as the largest eigenvalue of a 'deformed' Markov evolution operator [1] and, in one dimension, Bethe Ansatz allows its diagonalization. The large system-size limit can then be studied, as was done in some scaling regimes for the totally asymmetric exclusion process (TASEP) [73, 75], more generic asymmetric exclusion processes (ASEPs) [13], or the symmetric exclusion process (SSEP) [47].

In 1D, systems with a strong asymmetry (the difference between the left and right jump rates being e.g. of order 1, i.e. $L^0$, compared to the system size $L$) present a super-diffusive behavior of the Kardar–Parisi–Zhang (KPZ) [76] type: time has to be rescaled by a factor $L^z$, with a dynamical exponent $z = \frac{3}{2}$, in order for the macroscopic limit to be well-determined. On the other hand, symmetric systems or mildly asymmetric ones (such as the weakly asymmetric exclusion process, WASEP, where the difference of jump rates is of order $1/L$) are diffusive: they belong to the Edwards–Wilkinson (EW) class where now $t \sim L^z$ with $z = 2$. Such diffusive systems can also be studied using the Macroscopic Fluctuation Theory (MFT; see [77] for a review).

The time-averaged observables of interest are the average of the current $Q$ (counting the total number of jumps to the right minus those to the left) and of the activity $K$ (counting their sum). The current is odd by time reversal while the activity is even (see [78] for a review on the role of such even quantities, also called frenesy). In this paper we are interested in two aspects of the LDFs: on a generic ground, we study the large deviations of current and activity in the SSEP and ASEP models, showing that there exists a correspondence allowing one to map the LDFs of $K$ in the SSEP to that of $Q$ in the ASEP, with well-chosen jump rates – and more generally to map the joint distribution of $K$ and $Q$ between two ASEPs with different jump rates. We show that this correspondence is valid for systems with periodic boundary conditions and for open systems in contact with reservoirs of particles. On a more specific side, we also consider the large deviations of the activity $K$ in the SSEP: while it is known that such large deviations present a diffusive scaling regime in which a DPT takes place [47] and a hyperuniform phase at large activity [53], little is known on the scaling with $L$ apart from the diffusive regime. We show that one side of the transition to the hyperuniform regime is governed by a KPZ scaling phase, beyond the diffusive scalings. This comes perhaps as a surprise because both the dynamics of the SSEP and the observable $K$ are symmetric by time reversal.

The paper is organized as follows. In Sec. 2, we establish the mapping for spatially periodic systems, generalizing that of Ref. [53] and discussing how it can be implemented for complex values of the Lagrange multipliers associated with $K$ and $Q$ (with details gathered in Appendix A). In Sec. 3, we study a consequence of this mapping for the large deviations of activity in the SSEP, by mapping these to the large deviations of the current in an ASEP. We infer from this mapping an almost complete picture of the large- but finite-$L$ scaling governing the LDFs and DPT of the distribution of $K$ in the SSEP; we show in particular that deviations beyond the diffusive (i.e. EW) scalings, are governed by KPZ scalings on one side of the DPT. We show how the exponent of the structure factor that characterizes hyperuniformity is very robust and valid both in the EW and in the KPZ regimes. From this discussion, we also infer a conjecture on the asymptotic behavior of the XXZ spin chain two-point correlation function at fixed magnetization. In Sec. 4, we generalize the mapping to the case of open systems, for systems with boundary reservoirs. In particular, we show that the particle-hole symmetry breaking that occurs for the current LDF in the WASEP model can be mapped to a particle-hole symmetry breaking for the activity LDF, in another WASEP model.

# 2 Mapping joint activity and current fluctuations

## 2.1 SSEP, ASEP and observables of interest

The configurations $\mathcal{C}$ of the exclusion process we are interested in are given by the occupations (0 or 1 particle) on every site of a 1D chain of $L$ sites. We will consider both periodic boundary conditions (where site $L+1$ is identified with site 1) and open boundary conditions, for which particles can jump in and out of the system at fixed rates, at each extremity (sites 1 and $L$) – modeling the contact with infinite reservoirs. Particle jumps can occur only if the target site is empty, and:

- for the SSEP, jump rates are equal to 1;
- for the ASEP, particles jump to the right with rate $p$ and to the left with rate $q$.

To study large deviations, the joint probability distribution $P(\mathcal{C}, K, Q, t)$ to observe the system in a configuration $\mathcal{C}$ at time $t$ and values $K$ and $Q$ of the current and activity on the time window $[0, t]$ is Laplace transformed to

$$\hat{P}(\mathcal{C}, s, \mu, t) = \sum_{K \in \mathbb{Z}^+} \sum_{Q \in \mathbb{Z}} e^{sK + \mu Q} P(\mathcal{C}, K, Q, t). \tag{2.1}$$

One can show using standard methods (see e.g. [79]) that the time evolution of $\hat{P}$ is governed by a biased operator of evolution that depends on the "counters" (or Lagrange multipliers) $s$ and $\mu$, conjugated to $K$ and $Q$. Such passage from the 'microcanonical' ensemble, where $K$ and $Q$ are conditioned to take fixed values, to a 'canonical' ensemble, where the probability is exponentially biased as in (2.1), allows one to reconstitute the joint probability $P(K, Q, t) = \sum_{\mathcal{C}} P(\mathcal{C}, K, Q, t)$ in the large-time asymptotics. To do so, one considers the scaled cumulant generating function (SCGF), defined as

$$\psi(\mu, s) = \lim_{t \to \infty} \frac{1}{t} \log \left\langle e^{sK + \mu Q} \right\rangle = \lim_{t \to \infty} \frac{1}{t} \log \sum_{\mathcal{C}} \hat{P}(\mathcal{C}, s, \mu, t). \tag{2.2}$$

The joint probability density of $K$ and $Q$, $P(K, Q, t) = \sum_{\mathcal{C}} P(\mathcal{C}, K, Q, t)$, takes a large-deviation scaling form $\lim_{t \to \infty} \frac{1}{t} \log P(K = \mathrm{k}t, Q = \mathrm{q}t, t) = I(\mathrm{k}, \mathrm{q})$ where the rate function $I(\mathrm{k}, \mathrm{q})$ can be reconstituted from the SCGF $\psi(\mu, s)$ by a Legendre transform (see e.g. [80]). In this article, we focus on the biased ensemble where $s$ and $\mu$ are fixed. For exclusion processes, one can show that the evolution operator governing the time evolution of $\hat{P}$ is an XYZ spin chain with spin $\frac{1}{2}$, as we detail in Sec. 2.2.

## 2.2 Simple start: relating SSEP with ASEP

Using standard approaches (see e.g. [13]), in the SSEP with periodic boundary conditions ($L+1 \equiv 1$), the deformed Markov matrix associated with the activity can be written as a sum of local operators $m_{k,k+1}$ that describe the exchange between sites $k$ and $k+1$:

$$M = \sum_{k=1}^{L} m_{k,k+1} \quad \text{with} \quad m = \begin{pmatrix} 0 & 0 & 0 & 0 \\ 0 & -1 & e^s & 0 \\ 0 & e^s & -1 & 0 \\ 0 & 0 & 0 & 0 \end{pmatrix} = \frac{1}{2} \left( e^s (\sigma^x \otimes \sigma^x + \sigma^y \otimes \sigma^y) + \sigma^z \otimes \sigma^z - \mathbb{I} \otimes \mathbb{I} \right). \tag{2.3}$$

Here $\sigma^{x,y,z}$ are Pauli matrices. Note that $m$ corresponds to the XXZ local Hamiltonian (up to a global normalization by $e^s/2$ and a shift proportional to the identity), see a more detailed discussion in section 3.6. The dominant eigenvalue $\psi_K^{\text{SSEP}}(s)$ of this matrix is equal to the SCGF

defined in Eq. (2.2), for the activity only ($\mu = 0$). Using the above representation, one remarks that for $1 \le k \le L-1$

$$m_{k,k+1} + \frac{q-p}{2(p+q)}(\sigma_{k+1}^z - \sigma_k^z) = \frac{2}{p+q} w_{k,k+1} \quad \text{with} \quad w = \begin{pmatrix} 0 & 0 & 0 & 0 \\ 0 & -q & pe^{\mu} & 0 \\ 0 & qe^{-\mu} & -p & 0 \\ 0 & 0 & 0 & 0 \end{pmatrix}, \quad (2.4)$$

provided we have the following relations:

$$e^{2\mu} = \frac{q}{p}, \quad e^{2s} = \frac{4pq}{(p+q)^2} . \quad (2.5)$$

Note that these relations imply

$$s = -\log \cosh \mu \le 0 \quad (2.6)$$

as long as $s$ and $\mu$ are real parameters.

One observes that $w$ is the local ASEP Markov matrix, now with $\mu$ conjugated to the current $Q$ (and no counter for the activity). Altogether this leads to

$$M = \frac{2}{p+q} \sum_{k=1}^{L} w_{k,k+1} \equiv \frac{2}{p+q} W, \quad (2.7)$$

where $W$ is the deformed Markov matrix that counts the current in the ASEP with periodic boundary conditions. We denote by $\psi_Q^{\text{ASEP}}(\mu)$ its dominant eigenvalue, equal to the cumulant generating function of the particle current. Thus, we get the following connection between cumulant generating functions:

$$\psi_K^{\text{SSEP}}(s) = \frac{2}{p+q} \psi_Q^{\text{ASEP}}(\mu) \quad \text{with the relations} \quad e^{2s} = \frac{4pq}{(p+q)^2}, \quad e^{2\mu} = \frac{q}{p} . \quad (2.8)$$

In periodic boundary conditions, the total number of particles $N$ is conserved, so that this relation holds in each sector of constant $N$.

As we noted in Eq. (2.6), probing real values of the Lagrange multiplier $\mu$ associated with $Q$ in the ASEP only gives access to the negative range of values of the multiplier $s$ associated with $K$ in the SSEP. However, the relation (2.8) in fact *still holds* if $s \ge 0$ and $\mu$ is imaginary:

$$\psi_K^{\text{SSEP}}(s) = \frac{2}{p+q} \psi_Q^{\text{ASEP}}(\mu) \quad \text{with, for } s \ge 0: \quad \mu = i\mu', \quad \cos \mu' = e^{-s} \le 1, \quad \frac{q}{p} = e^{2i\mu'} . \quad (2.9)$$

In that case, as we show in Appendix A, $\psi_Q^{\text{ASEP}}(\mu)$ has to be understood as the unique branch of the spectrum of the finite-size matrix $W$ which goes to 0 as $\mu \to 0$ (with $\mu$ seen as a complex number), which is well defined even if $p, q \in \mathbb{C}$, as in our case. For $\mu, p, q \in \mathbb{R}$, such definition of $\psi_Q^{\text{ASEP}}(\mu)$ coincides with the eigenvalue of $W$ with the largest real part; while for complex $\mu, p, q$ verifying Eq. (2.9) we show that $\psi_Q^{\text{ASEP}}(i\mu')$ is analytical for $\mu' \in ]-\frac{\pi}{2}, \frac{\pi}{2}[$ (corresponding to $0 < e^{-s} \le 1$). This formal remark will allow us to use the results obtained by Prolhac in Ref. [81] since he precisely identifies $\psi_Q^{\text{ASEP}}(\mu)$ through Bethe Ansatz using the criterion that it is the unique branch of the spectrum of $W$ that goes to 0 as $\mu \to 0$ (see Appendix A for details). Note that Jack, Thompson and Sollich, in Ref. [53], obtained a mapping equivalent to Eq. (2.8) but used it only in the range $s < 0$ and $p, q > 0$, while in the present paper we study in Sec. 3.3 the regime of Eq. (2.9) in order to gain physical insight on the large-activity deviations, and to show that the hyperuniform phase is surprisingly difficult to study beyond the diffusive regime.

## 2.3  Relation between two periodic ASEPs

One can generalize the previous computation to a mapping between two ASEP models with counters for both the current and the activity. We denote by $p_i$, $q_i$ the jump rates of these ASEP models and by $s_i, \mu_i$ the counters associated with the activity and current, with $i = 1, 2$. The local operator are:

$$w(s_1,\mu_1) = \begin{pmatrix} 0 & 0 & 0 & 0 \\ 0 & -q_1 & p_1 e^{s_1+\mu_1} & 0 \\ 0 & q_1 e^{s_1-\mu_1} & -p_1 & 0 \\ 0 & 0 & 0 & 0 \end{pmatrix} \quad \text{and} \quad m(s_2,\mu_2) = \begin{pmatrix} 0 & 0 & 0 & 0 \\ 0 & -q_2 & p_2 e^{s_2+\mu_2} & 0 \\ 0 & q_2 e^{s_2-\mu_2} & -p_2 & 0 \\ 0 & 0 & 0 & 0 \end{pmatrix},$$

(2.10)

corresponding to the deformed Markov matrices (with periodic boundary conditions $L+1 \equiv 1$)

$$W = \sum_{k=1}^{L} w_{k,k+1}(s_1,\mu_1) \quad \text{and} \quad M = \sum_{k=1}^{L} m_{k,k+1}(s_2,\mu_2).$$

(2.11)

Then we have

$$m_{k,k+1}(s_2,\mu_2) = \frac{p_2+q_2}{p_1+q_1} w_{k,k+1}(s_1,\mu_1) + \frac{p_1 q_2 - p_2 q_1}{2(p_1+q_1)} \left( \sigma_k^z - \sigma_{k+1}^z \right),$$

(2.12)

provided the following relations are verified:

$$\mu_1 - \mathcal{E}_1 = \mu_2 - \mathcal{E}_2 \quad \text{with} \quad \mathcal{E}_j = \log\sqrt{\frac{q_j}{p_j}}, \ j = 1, 2$$

$$s_1 - \log\cosh(\mathcal{E}_1) = s_2 - \log\cosh(\mathcal{E}_2).$$

(2.13)

This implies $M = \frac{p_1+q_1}{p_2+q_2} W$. Denoting the SCGF of Eq. (2.2) by writing the rates explicitly as $\psi(\mu, s | p, q)$, the computation above has lead to the following property for the joint cumulant generating function:

$$\psi(\mu_1, s_1 | p_1, q_1) = \frac{p_1+q_1}{p_2+q_2} \psi\left(\mu_1 - \frac{1}{2}\log\frac{q_1 p_2}{p_1 q_2}, s_1 - \log\frac{\sqrt{p_2 q_2}(p_1+q_1)}{\sqrt{p_1 q_1}(p_2+q_2)} \Big| p_2, q_2 \right).$$

(2.14)

One recovers the results of the previous section for $p_1 = q_1 = 1$, $s_2 = 0$, $\mu_1 = 0$.

The correspondence (2.14) can also be reexpressed using the fact that, as seen from Eq. (2.10) the SCGF $\psi(\mu, s | p, q)$, once factorized by the global rate $p + q$, is a function of $\mathcal{E} = \log\sqrt{q/p}$ only:

$$\psi(\mu, s | p, q) = (p + q)\mathring{\psi}(\mu, s | \mathcal{E}),$$

(2.15)

with then

$$\mathring{\psi}(\mu_1, s_1 | \mathcal{E}_1) = \mathring{\psi}\left(\mu_1 - \mathcal{E}_1 + \mathcal{E}_2, s_1 - \log\frac{\cosh\mathcal{E}_1}{\cosh\mathcal{E}_2} \Big| \mathcal{E}_2 \right).$$

(2.16)

**Gallavotti-Cohen symmetry point:**  In the present context, the Gallavotti-Cohen (GC) symmetry [82–85] of the ASEP is expressed as $\psi(\mu, s | p, q) = \psi(2\mathcal{E} - \mu, s | p, q)$, meaning that the function $\psi$ is symmetric around $\mathcal{E}$ in terms of its argument $\mu$. Considering the specific case of the mapping between the activity in the SSEP ($p_1 = q_1 = 1$) and the current in the ASEP, one has $p_1 = q_1 = 1$, $s_2 = 0$, $\mu_1 = 0$, which implies that $\mu_2 = \mathcal{E}_2$. In this case, the SSEP is mapped on the ASEP precisely at its Gallavotti-Cohen's symmetry point $\mu_2 = \mathcal{E}_2$, and applying the GC symmetry simply maps the ASEP to itself. Now considering the more general mapping (2.14), yet specified to the SSEP for system 1 by setting $p_1 = q_1 = 1$, but leaving $\mu_1$ nonzero, one sees that the ASEP obtained by the mapping has $\mu_2 = \mu_1 + \mathcal{E}_2$, which differs from the GC

symmetry point $\mu_2 = \mathcal{E}_2$. By applying the GC symmetry to the ASEP and mapping back to the SSEP, we eventually obtain $\psi(\mu_1, s_1 | 1, 1) = \psi(-\mu_1, s_1 | 1, 1)$, that is simply the time (or space) reversal symmetry of the SSEP. More generally, the mapping between two ASEPs maps the GC symmetry of an ASEP to the GC symmetry of the other ASEP.

## 2.4   Relation with macroscopic fluctuation theory

The above correspondence of Eq. (2.14) can be understood in the MFT framework, by considering the WASEP limit where left and right jump rates scale as

$$p = e^{\frac{\nu}{L}}, \quad q = e^{-\frac{\nu}{L}}. \tag{2.17}$$

Within MFT, the probability to observe given density and current profiles $\rho(x, t)$ and $j(x, t)$, related by the continuity equation $\partial_t \rho + \partial_x j = 0$, is expressed as

$$P[\rho, j] \propto e^{-L\mathcal{A}[\rho, j]} \quad \text{with} \quad \mathcal{A}[\rho, j] = \int_0^T dt \int_0^1 dx \frac{\left[j + \partial_x \rho - 2\nu \rho(1-\rho)\right]^2}{4\rho(1-\rho)}, \tag{2.18}$$

where the macroscopic limit of the lattice model has been taken within the diffusive scaling $x = i/L \in [0,1]$ and $T = t/L^2$. Remarking the identity

$$\mathcal{A}[\rho, j] = \int_0^T dt \int_0^1 dx \left\{ \frac{[j + \partial_x \rho]^2}{4\rho(1-\rho)} - \nu j + \nu^2 \rho(1-\rho) \right\}, \tag{2.19}$$

where we have used the periodic boundary condition $\rho(1, t) = \rho(0, t)$, we obtain the following relation between the actions of two different WASEP models with asymmetry parameters $\nu_1$ and $\nu_2$:

$$\mathcal{A}_2[\rho, j] = \mathcal{A}_1[\rho, j] + (\nu_1 - \nu_2) \int_0^T dt \int_0^1 dx\, j + (\nu_2^2 - \nu_1^2) \int_0^T dt \int_0^1 dx\, \rho(1-\rho). \tag{2.20}$$

In the MFT framework, the activity and integrated current read [47]

$$K = 2L^3 \int_0^T dt \int_0^1 dx\, \rho(1-\rho) \quad \text{and} \quad Q = L^2 \int_0^T dt \int_0^1 dx\, j \tag{2.21}$$

and the joint cumulant generating function is obtained through minimization of

$$\int \mathcal{D}j \int \mathcal{D}\rho\, \exp\left(-L\mathcal{A}[\rho, j] + \mu Q + sK\right). \tag{2.22}$$

Using relation (2.20), we infer the conditions

$$\mu_1 + \frac{\nu_1}{L} = \mu_2 + \frac{\nu_2}{L} \quad \text{and} \quad s_1 - \frac{\nu_1^2}{2L^2} = s_2 - \frac{\nu_2^2}{2L^2}, \tag{2.23}$$

which reproduce (2.13) in the WASEP limit (2.17). We thus have shown that the generic relation (2.13) valid for any finite system size is compatible with results obtained in the diffusive scaling through the MFT path-integral representation of the trajectory probability.

# 3 Large-activity fluctuations in the periodic SSEP

## 3.1 Context and motivation

Large deviations of the distribution of activity in the SSEP were studied in [47], in the large system-size limit $L \to \infty$ and within a diffusive scaling[1] that allows one to use MFT [86]. Such regime also corresponds to the Edwards–Wilkinson (EW) type of fluctuations. In such diffusive scaling limit, the Lagrange multiplier $s$ for the activity scales as $s = \lambda/L^2$ in order for the large deviations to remain diffusive. This can be seen from the expressions (2.21) of the activity $K$ and the path-integral representation (2.22): choosing $s = \lambda/L^2$ allows one to get the same prefactor $L$ in front of the action term and of the activity term. In periodic systems, it was shown in [47] that in the small activity regime ($\lambda < 0$) a phase transition occurs at a finite value $\lambda_c < 0$, which marks the occurrence of a symmetry breaking: for $\lambda \geq \lambda_c$, the typical particle density profile is flat, while for $\lambda < \lambda_c$ the density profile is not translation-invariant anymore (it presents a kink and an anti-kink). Such transitions express the fact that a lower activity induces a clustering of particles. The optimal profile can be computed [51] and, in the very inactive phase $\lambda \to -\infty$, the system becomes fully phase separated. Such a transition between a flat profile and non-flat profiles is reminiscent of the one observed in the large deviations of the current in the WASEP [43, 45] where a symmetry breaking also occurs: the flat optimal profile for current deviations close enough to the mean current abruptly transforms into a traveling inhomogeneous profile when the deviations of current are large enough.

The opposite regime of very large activity fluctuations (large positive $\lambda$ or $s$) was studied in [53, 87]. For $s \to \infty$, the biased operator is dominated by its non-diagonal part and a Jordan–Wigner transformation allows one to identify the dominant scaling of the SCGF [87]. For intermediate values of $s > 0$, a hyperuniform behavior was found in Ref. [53] where it was also argued that a phase transition occurs for $s = 0$ between a phase-separated regime at $s < 0$ and a hyperuniform regime at $s > 0$. Physically, the hyperuniformity describes a regime where particles develop an effective repulsion in order to increase their propensity to keep empty sites around them, which favors possible jumps[2] (and thus trajectories with higher activity).

Here, thanks to a detailed analysis of the mapping established in Sec. 2, we provide a more detailed analysis of how the transition through the various regimes of activity fluctuations occurs as $s$ is varied. We find that the transition around $s = 0$ is governed by KPZ-type scalings on one side of the transition. This comes as a surprise since KPZ fluctuations are often associated with a breaking of reversibility (as in the KPZ equation itself or in current fluctuations of asymmetric exclusion processes) while, in our case of interest, the activity fluctuations of the SSEP do not break the time-reversal symmetry.

Summarizing the results of previous works and those of the present paper, we find the following picture, from very small activity to very large activity (denoting $\rho$ the average occupation in the system):

I) **Very small activity ($s \to -\infty$)** [51,87]: $\psi_K^{\mathrm{SSEP}}(s) = -4L\rho(1-\rho)(1-e^s) + O(e^{2s})$.
In this regime, the eigenstate is the one which minimizes the escape rate: it is degenerate and corresponds to configurations where all particles are gathered in a single cluster. The result for the SCGF is obtained by a perturbation in $e^s \ll 1$.

II) **Around $s = 0$: diffusive regime $s = \lambda/L^2$ with $\lambda = O(L^0)$** [*i.e.* $s \to 0$, $L \to \infty$ with $\lambda = sL^2$ fixed]

---

[1]In such regime, the site index $i$ becomes a continuous coordinate $x = i/L$ with $0 \leq x \leq 1$, while the microscopic time scale $t$ becomes a macroscopic one $T = t/L^2$.

[2]Strictly speaking this physical picture is valid only for a density less than $1/2$ ; for a larger density, the roles of particles and holes need to be exchanged.

○ *Small activity ($\lambda \to -\infty$)* [51]: for $\rho = \frac{1}{2}$,

$$\psi_K^{\text{SSEP}}(s) = -\frac{2}{L}\sqrt{2|\lambda|} - \frac{8}{L}\sqrt{2|\lambda|}e^{-\sqrt{|\lambda|/2}} + \frac{1}{L}O(\lambda e^{-\sqrt{2|\lambda|}}). \tag{3.1}$$

○ *Symmetry breaking at $\lambda_c = -\frac{\pi^2}{2\rho(1-\rho)} < 0$*: for $\lambda < \lambda_c$, each possible optimal profile breaks the translational invariance by presenting a kink and an antikink [51].

○ *Diffusive fluctuations for $\lambda_c < \lambda < \infty$*, described by a universal function $\mathcal{F}(u)$ [47]:

$$\psi_K^{\text{SSEP}}(s) = \frac{\lambda}{L^2}\frac{\langle K \rangle}{t} + \frac{1}{L^2}\mathcal{F}\big(-\rho(1-\rho)\lambda\big) + o(L^{-2}). \tag{3.2}$$

○ *Large activity fluctuations ($\lambda \to \infty$)*, in the diffusive (EW) regime (see Sec. 3.4) [47]:

$$\psi_K^{\text{SSEP}}(s) - s\frac{\langle K \rangle}{t} \underset{\lambda = sL^2 \to \infty}{\sim} \frac{4}{3\pi}\frac{1}{L^2}\big[2\rho(1-\rho)\big]^{\frac{3}{2}}\lambda^{\frac{3}{2}}. \tag{3.3}$$

III) **In a KPZ-type rescaling $s = O(L^0)$, one finds (see Sec. 3.3):**

$$\psi_K^{\text{SSEP}}(s) = 2L\rho(1-\rho)f(s) + \mathcal{K}(s) \quad \text{for } s \leq 0, \tag{3.4}$$

where the functions $f(s)$ and $\mathcal{K}(s)$ are determined thanks to the mapping presented in Sec. 2.2, using the results of [81, 88]. In the $s \to 0^-$ asymptotics of (3.4) one finds

$$\psi_K^{\text{SSEP}}(s) - s\frac{\langle K \rangle}{t} \underset{s \to 0^-}{\sim} \sqrt{2\pi}L^{3/2}\big[\rho(1-\rho)\big]^{3/2}(-s)^{3/2}. \tag{3.5}$$

We were not able to extract a similar asymptotic behavior for the regime $s \to 0^+$. We describe in Sec. 3.5 the origin of the issues at stake.

IV) **Very large activity fluctuations $s \to \infty$ [87]:**

$$\frac{1}{L}\psi_K^{\text{SSEP}}(s) \underset{s \to +\infty}{\sim} 2e^s\frac{\sin \pi\rho}{\pi} - 2\rho(1-\rho) - 2\frac{\sin^2 \pi\rho}{\pi^2} + O(e^{-s}). \tag{3.6}$$

That phase, at dominant order (proportional to $e^s$), is effectively described by a XY Hamiltonian (see the operator of Eq. (2.3)). A Jordan–Wigner transformation allows one to identify the dominant scaling, and the next term, of order $(e^s)^0$, is obtained by perturbation theory. This phase is far from the MFT regime (in the sense that the lattice structure dominates the physics).

In other words, in the present paper, we show that the phase-separation / hyperuniform transition described in [53] when varying $s$ by finite amount $s = O(L^0)$ is in fact described by a well-determined transition at large but finite $L$ for $s = \lambda/L^2$ with a transition point at $\lambda = \lambda_c < 0$ in the diffusive EW regime. The approach to the transition for $s \to 0^-$ is governed by a singular power-law behaviour of the form $\psi_K^{\text{SSEP}}(s) - s\frac{1}{t}\langle K \rangle \propto L^{3/2}|s|^{3/2}$, whose exponent is inherited from KPZ scalings. We also discuss in Sec. 3.7 the power-law behavior of the structure factor as a function of the wave number.

## 3.2 Diffusive regime: mapping between the fluctuations of activity in SSEP and current in WASEP

We consider the special case where $p_1 = q_1 = 1$ and $\mu_1 = 0$, $s_2 = 0$. It allows one to map the activity fluctuations in a SSEP to the current fluctuations in an ASEP. For sake of simplicity, and since there is no possible confusion, we drop the indices and call $s$ the counter of activity $K$ in

the SSEP, $\mu$ the counter (or fugacity) of the current $Q$ in the ASEP, and $p$, $q$ the jump rates of the ASEP. The above construction then reads

$$\psi_K^{\text{SSEP}}(s) = \frac{2}{p+q}\psi_Q^{\text{ASEP}}(\mu) \quad \text{with} \quad e^{-2\mu} = \frac{p}{q} \quad \text{and} \quad e^{-s} = \cosh(\mu), \tag{3.7}$$

where, see Sec. 2.2 and Appendix A, $\mu \in \mathbb{R}$ for $s \leq 0$ and $\mu \in i\,]-\frac{\pi}{2},\frac{\pi}{2}[$ for $s > 0$.

In this subsection, we focus on the diffusive (or Edwards–Wilkinson) regimes of fluctuations. In the SSEP, these describe the activity fluctuations in the regime $s = \lambda/L^2$ with fixed $\lambda$ as $L \to \infty$ [47]. Using the mapping (3.7), we see that this regime corresponds for the ASEP to current fluctuations in the regime $\mu = \kappa/L$ with fixed $\kappa$ as $L \to \infty$, and to a weak asymmetry $p - q = O(1/L)$ (i.e. to the so-called WASEP). In fact, for $s > 0$, as detailed in Eq. (2.9), the WASEP model has complex rates such that $p/q = e^{-2i\mu'}$ with $e^{-s} = \cos\mu'$. Yet, as discussed in Appendix A, the maximal eigenvalue of the evolution operator is the one which goes to 0 as $\mu' \to 0$, which allows one to use the result of Prolhac [81], who showed that the cumulant of order $n$ of the current is

$$\frac{1}{t}\langle Q^n \rangle_c^{\text{WASEP}} = p\sum_{j=\lceil n/2 \rceil}^{n}\binom{j}{n-j}\frac{n!B_{2j-2}}{j!(j-1)!}\Big(1-\frac{q}{p}\Big)^{2j-n}\big[\rho(1-\rho)\big]^j L^{2j} \qquad (n \geq 3), \tag{3.8}$$

where $B_{2j-2}$ are Bernoulli numbers $B_2 = \frac{1}{6}$, $B_4 = -\frac{1}{30}$,... A tedious but straightforward computation shows that the mapping (3.7) implies that the cumulants of the activity in the SSEP take a simpler expression

$$\frac{1}{t}\langle K^n \rangle_c^{\text{SSEP}} = \frac{B_{2n-2}}{(n-1)!}\big[2\rho(1-\rho)\big]^n L^{2n-2} \qquad (n \geq 2). \tag{3.9}$$

As expected by consistency, this expression is exactly the one obtained through MFT and coordinate Bethe Ansatz in Ref. [47].

This shows that the mapping (3.7), in the diffusive regime, works both for $s \leq 0$ (where $p, q, \mu \in \mathbb{R}$) and $s > 0$ (where $p, q, \mu \in \mathbb{C}$), and this because the SCGF we considered are analytical in a vicinity of 0 in that regime.

## 3.3 Application of the mapping: a KPZ scaling in the distribution of activity of the SSEP

To go beyond the diffusive regime, we now detail the regime III announced above, that corresponds to $s = O(L^0)$ for the fluctuations of activity in the SSEP and to $\mu = O(L^0)$ for the fluctuations of current in the ASEP with a finite non-zero asymmetry $p - q$.

In that regime, the SCGF $\psi_Q^{\text{ASEP}}(\mu)$ has been computed by Bethe Ansatz (see equation (11) in Ref. [88] by Lee and Kim, or equation (117) in Ref. [81] by Prolhac), in the thermodynamic limit $L \to \infty$, and can be expressed parametrically as:

$$\psi_Q^{\text{ASEP}}(\mu) - \bar{\jmath}L\mu = -(p-q)\sqrt{\frac{\rho(1-\rho)}{2\pi L^3}}\,\text{Li}_{5/2}(B) = \sum_{k=2}^{\infty}E_k\frac{\mu^k}{k!} \tag{3.10}$$

$$\mu = \frac{-1}{\rho(1-\rho)}\sqrt{\frac{\rho(1-\rho)}{2\pi L^3}}\,\text{Li}_{3/2}(B), \tag{3.11}$$

with $\bar{\jmath} = (p-q)\rho(1-\rho) = \lim_{L\to\infty}\lim_{t\to\infty}\frac{\langle Q \rangle}{Lt}$ being the average current[3]. We have used the following convention for polylogarithm functions: $\text{Li}_r(x) = \sum_{k=1}^{\infty}\frac{x^k}{k^r}$. The above expres-

---

[3]Note that if, in the left-hand side of Eq. (3.10), one subtracts $J\mu$ with $J = \lim_{t\to\infty}\frac{\langle Q \rangle}{t} = (p-q)\rho(1-\rho)\frac{L^2}{L-1}$ instead of subtracting $\bar{\jmath}L\mu$, one can write (3.10) with an extra $\text{Li}_{3/2}$ function on the right-hand side, coming from the expansion $\frac{L^2}{L-1} = L + 1 + O(L^{-1})$ and the use of (3.11). This is the convention taken for instance in equation (117) in Ref. [81]. See also the discussion after equation (24) in Ref. [88].

sions have to be understood as series expansions in the auxiliary variable $B$ that allow one to reconstruct the series in $\mu$. The expressions (3.10) and (3.11) are valid as long as $p-q$ is finite (i.e. of order $L^0$) and $L$ is sent to infinity, see §6.3 in Ref. [81]. The cumulants $E_k$ of the current defined in the series expansion in powers of $\mu$ in (3.10) are also defined in that limit.

To illustrate how the parametric representation of Eqs. (3.10)-(3.11) works, we compute the two first cumulants. Expanding in powers of $B$, one has:

$$\mu = \frac{-\bar{a}}{\rho(1-\rho)}\left(B + \frac{\sqrt{2}}{4}B^2 + ...\right) \quad \text{with} \quad \bar{a} = \sqrt{\frac{\rho(1-\rho)}{2\pi L^3}} \tag{3.12}$$

$$\psi_Q^{\text{ASEP}}(\mu) = \bar{j}L\mu + (p-q)\bar{a}\left(B + \frac{\sqrt{2}}{8}B^2 + ...\right) \tag{3.13}$$

$$= E_1\mu + E_2\frac{\mu^2}{2!} + ... \tag{3.14}$$

Substituting (3.12) into (3.13) and (3.14) and equating the series in powers of $B$ leads to

$$E_1 = \bar{j}(L+1) = (p-q)(L+1)\rho(1-\rho) \quad \text{and} \quad E_2 = \frac{\sqrt{\pi}}{2}(p-q)[\rho(1-\rho)]^{3/2}L^{3/2}. \tag{3.15}$$

The expression of $E_2$ is in fact valid for $p-q \gg \frac{1}{\sqrt{L}}$ (and not only for non-zero $p-q = O(L^0)$, see §2.3 in Ref. [81]).

For the SCGF of activity in the SSEP, plugging (3.7) into (3.10)-(3.11), we get, for $s \le 0$:

$$\psi_K^{\text{SSEP}}(s) = -2\bar{a}\left(L\,\text{Li}_{3/2}(B) + \text{Li}_{5/2}(B)\right)\tanh\left(\frac{\bar{a}}{\rho(1-\rho)}\text{Li}_{3/2}(B)\right) \tag{3.16}$$

$$e^{-s} = \cosh\left(\frac{\bar{a}}{\rho(1-\rho)}\text{Li}_{3/2}(B)\right). \tag{3.17}$$

This result can be rewritten as

$$\psi_K^{\text{SSEP}}(s) = -2L\rho(1-\rho)f(s) + \mathcal{K}(s), \tag{3.18}$$

with, again for $s \le 0$:

$$f(s) = \sqrt{1-e^{2s}}\,\text{arccosh}\left(e^{-s}\right) = \sum_{n=0}^{\infty}\frac{(1-e^{2s})^{n+1}}{2n+1} \tag{3.19}$$

$$\mathcal{K}(s) = -2\bar{a}\,\text{Li}_{5/2}(B)\tanh\left(\frac{\bar{a}}{\rho(1-\rho)}\text{Li}_{3/2}(B)\right) \tag{3.20}$$

$$s = -\log\left[\cosh\left(\frac{\bar{a}}{\rho(1-\rho)}\text{Li}_{3/2}(B)\right)\right], \tag{3.21}$$

with $\bar{a}$ given in (3.12). Note that $f(s)$ is an analytical function of $s \in \mathbb{R}$ and can be expanded as a power series in $s$. The expressions (3.18)-(3.21) (valid for $s \le 0$) are the equivalent for the SCGF of the activity in the SSEP of the parametric expression of the SCGF of the current in the ASEP given by Eqs. (3.10)-(3.11).

Solving iteratively the system of equations (3.20)-(3.21) in order to eliminate $B$, we get from (3.21) that at minimal order $B = \frac{\sqrt{-2s}}{\bar{a}}\rho(1-\rho) + o(\sqrt{-s})$. This means that, in contrast to $f(s)$, the function $\mathcal{K}(s)$ has to be expanded as a series of $\sqrt{-s}$ as $\mathcal{K}(s) = \sum_{n=1}^{\infty}\mathcal{K}_n\frac{(\sqrt{-s})^n}{n!}$. From (3.20), we obtain that $\mathcal{K}(s) = \bar{a}^2B^3/[2\sqrt{2}\rho(1-\rho)] + o(B^3)$. Thus, for the first orders of the series expansion of $\mathcal{K}(s)$, one obtains: $\mathcal{K}_1 = \mathcal{K}_2 = 0$ and an explicit expression of $\mathcal{K}_3$ which, gathering the results, implies finally that as $s \to 0^-$:

$$\psi_K^{\text{SSEP}}(s) - 2L\rho(1-\rho)s = \mathcal{K}_3\frac{(-s)^{\frac{3}{2}}}{3!} + O(s^2) \quad (s \le 0) \tag{3.22}$$

$$\mathcal{K}_3 = 6\sqrt{2\pi}L^{3/2}\left[\rho(1-\rho)\right]^{3/2}. \tag{3.23}$$

Such expression is valid in the 'KPZ regime' described by Prolhac in Ref. [81], corresponding to $p - q \gg \frac{1}{\sqrt{L}}$ for the ASEP model. Reading from (3.7), this corresponds for the SSEP to the condition $|s| \gg \frac{1}{L}$ (because, without numerical prefactors, $s \sim \mu^2$ and $\mu \sim \log \frac{p}{q}$).

Note that the computation above is valid for $s \leq 0$ since Eqs. (3.10)-(3.11) for $\psi_Q^{\text{ASEP}}(\mu)$ are valid for real values of $\mu$. Although the regime $s > 0$ corresponding to imaginary values of $\mu$ could in principle be obtained from $\psi_Q^{\text{ASEP}}(\mu)$ (see Appendix A), we cannot perform an analytical continuation of Eqs. (3.10)-(3.11) to complex values of $\mu, B, p, q$, as we detail in Sec. 3.5. This is related to the fact that although Eqs. (3.10)-(3.11) yield for $\psi_Q^{\text{ASEP}}(\mu)$ a power series in $\mu$, the mapping to $\psi_K^{\text{SSEP}}(s)$ yields a function which is non analytic in the vicinity of $s = 0$, as our result (3.22) shows.

The behavior (3.22) of the SCGF characterizes the KPZ nature of the fluctuations of activity in the SSEP, which are non-diffusive, and occur at a large-enough deviation of activity ($|s| \gg \frac{1}{L}$). To see how they are possibly connected to smaller fluctuations of activity, we now analyze in Sec. 3.4 the regime of diffusive (EW) fluctuations of activity.

### 3.4 Large-activity fluctuations of the Edwards–Wilkinson regime

Adapting the notations of [47] ($e^{-s}$ becomes $e^{+s}$), one finds that small fluctuations of activity (i.e in a scaling regime $s \to 0$, $L \to \infty$ with $\lambda = sL^2$ fixed) fall into the EW class, described by the MFT. The SCGF in that regime (where $s$ depends implicitly on $L$) reads

$$\psi_K^{\text{SSEP}}(s) = s\frac{\langle K \rangle}{t} + \frac{1}{L^2}\mathcal{F}\left(-\rho(1-\rho)L^2 s\right) + o(L^{-2}) \qquad \lambda = sL^2 \text{ fixed}, \tag{3.24}$$

with $\frac{\langle K \rangle}{t} = 2\rho(1-\rho)\frac{L^2}{L-1}$ and

$$\mathcal{F}(u) = \sum_{k \geq 2}\frac{B_{2k-2}}{(k-1)!k!}(-2u)^k \tag{3.25}$$

$$\overset{(u<0)}{=} 2|u| + \frac{4\sqrt{2}}{\pi}|u|^{\frac{3}{2}}\int_{-1}^{1} dy\, y^2 \coth\left(\sqrt{2|u|}\sqrt{1-y^2}\right). \tag{3.26}$$

These three equations are rewritings of Eqs. (14), (17) and (A2) of [47], and the $B_{2k}$'s are Bernoulli numbers $B_2 = \frac{1}{6}$, $B_4 = -\frac{1}{30}$, ... The large $\lambda$ regime corresponds physically to a regime of very large values of $K$, and mathematically to the asymptotics $u \to -\infty$ with $u = -\rho(1-\rho)L^2 s$. In that regime, the coth in (3.26) can be safely replaced by 1 (the boundary layer around $|y| = 1$ has a lesser and lesser contribution as $u \to -\infty$) and (3.26) gives

$$\mathcal{F}(u) \underset{u \to -\infty}{\sim} \frac{8\sqrt{2}}{3\pi}|u|^{\frac{3}{2}}. \tag{3.27}$$

Inserting into (3.24) one thus gets that in the EW diffusive scaling regime

$$\psi_K^{\text{SSEP}}(s) - s\frac{\langle K \rangle}{t} \underset{\substack{\lambda \to \infty \\ \lambda = sL^2}}{\sim} \frac{8\sqrt{2}}{3\pi}L\left[\rho(1-\rho)\right]^{\frac{3}{2}}s^{\frac{3}{2}}, \tag{3.28}$$

which is valid for $s \geq 0$ and $s = O(1/L^2)$. Compared to the KPZ-scaling-regime result of Eqs. (3.22)-(3.23), which is valid for $s \leq 0$ with $|s| \gg \frac{1}{L}$, we observe that the exponent of $|s|$ is the same, but that the prefactor (including the exponent of $L$) is different. This indicates that both sides of the transition around $s = 0$ behave in a different manner. Understanding how the SCGF behaves for $s > 0$ (beyond the diffusive regime) is not trivial and still an open question, as we detail in the next subsection.

## 3.5 The non-diffusive regime $s > 0$: difficulties and open questions

As explained in Sec. 2.2 and Appendix A, the mapping (3.7) between $\psi_Q^{\mathrm{ASEP}}(\mu)$ and $\psi_K^{\mathrm{SSEP}}(s)$ is valid for $\mu \in \mathbb{R}$ if $s \leq 0$ and $\mu \in i\,]-\frac{\pi}{2}, \frac{\pi}{2}[$ (with $\frac{q}{p} = e^{2\mu}$) if $s < 0$. As we now detail, using this mapping proves surprisingly difficult in the non-diffusive regime $s > 0$.

To explain this, we start from the exact expression of the diffusion constant $D = \lim_{t\to\infty} \frac{\langle Q^2\rangle_c}{t}$ for finite size $L$ and finite occupation $N$. It was obtained for the ASEP by a generalization of the matrix method [89] or by varied versions of the Bethe Ansatz [17,90,91], generalizing previous computations that were done for the TASEP [13,73,75], and it reads:

$$D = \frac{2(p-q)L}{L-1} \sum_{k=1}^{L} k^2 \frac{1+\left(\frac{q}{p}\right)^k}{1-\left(\frac{q}{p}\right)^k} \frac{\binom{L}{N+k}\binom{L}{N-k}}{\binom{L}{N}^2} \ . \tag{3.29}$$

Noting that $q/p = e^{2i\mu'}$ and increasing $s$ from 0 (which also makes $\mu'$ increase from 0), we see that the denominator of (3.29) becomes equal to zero first when $e^{2iL\mu'} = 1$. This yields a diverging diffusion coefficient for $\mu' \to \mu_c'^{-}$ with

$$\mu_c' = \frac{2\pi}{L}. \tag{3.30}$$

We have checked numerically the existence of such a divergence by diagonalizing the deformed matrix $W$ (for the parameters of the mapping of Eq. (3.7) taking complex values as in (2.9)), for values of $N$ and $L$ up to 16. The evaluation of the diffusion coefficient $D$ was performed by using the relation $D = \lim_{t\to\infty} \frac{\langle Q^2\rangle_c}{t}$. It implies that $D$ can be obtained from the second derivative of the SCGF of the current in 0. The SCGF was computed with arbitrary precision, allowing to take the second derivative numerically. The obtained value of $D$ diverges when increasing $\mu'$ close to $\mu_c'$.

To check that this divergence remains present in the thermodynamic limit, one can study the large-size behaviour of (3.29) following [91]. Using Stirling formula in the thermodynamic limit ($L, N \to \infty$ with $N/L \to \rho$)

$$\frac{\binom{L}{N\pm k}}{\binom{L}{N}} \sim \left[\frac{1-\rho}{\rho}\right]^{\pm k} e^{-\frac{k^2 \pm (1-2\rho)k}{2\rho(1-\rho)L}} \tag{3.31}$$

together with the condition $\frac{q}{p} = e^{2i\mu'}$ of Eq. (2.9), we obtain

$$D \sim 2(p-q) \sum_{k=1}^{L} \frac{ik^2}{\tan k\mu'} e^{-\frac{k^2}{\rho(1-\rho)L}} \ , \tag{3.32}$$

which shows that the divergence remains present in $\mu_c$ given by (2.9) as $\mu'$ increases from 0.

Cumulants of higher order have also been computed [81], and their expression also involves fractions of the form $1/\left(1-\left(\frac{q}{p}\right)^k\right)$ and they all diverge at the same point $\mu_c'$ given by Eq. (3.30). This means that the series

$$\frac{2}{p+q}\psi_Q^{\mathrm{ASEP}}(\mu) = \frac{2}{p+q} \sum_{n\geq 0} \frac{\langle Q^k\rangle_c}{t} \frac{\mu^k}{k!} \quad \text{with} \quad e^{-2\mu} = \frac{p}{q} \quad \text{and} \quad e^{-s} = \cosh(\mu)\,, \tag{3.33}$$

which is equal to $\psi_K^{\mathrm{SSEP}}(s)$, has all its coefficients diverging for $\mu = i\mu_c'$ but is an analytic function of $\mu$ in a vicinity of $\mu \in i\,]-\frac{\pi}{2}, \frac{\pi}{2}[$, as shown in Appendix A. We were not able to perform a resummation or a transformation of this series that would allow to cure this behavior.

A possibility could be to write $\frac{q}{p} = e^{2i\mu'-\varepsilon}$ with a small $\varepsilon > 0$. If one assumes that the limits $\varepsilon \to 0$ and $L \to \infty$ commute, one can (i) send first $L$ to $\infty$: since there are no more divergences in the cumulants as $\mu$ varies, one can use the same $L \to \infty$ asymptotics as in Ref. [81] and (ii) then send $\varepsilon$ to 0. One obtains the same parametric representation of $\psi_Q^{\text{ASEP}}(\mu)$ as in Eqs. (3.10)-(3.11), which were obtained by extracting the large-$L$ behavior of the cumulants of the current (see Ref. [81]). To determine $\psi_K^{\text{SSEP}}(s)$ from the mapping (3.7), one thus has to find the curve of locations where $\text{Li}_{3/2}(B)$ is purely imaginary in the complex plane, to solve Eq. (3.11). The natural cuts of this polylogarithm function intersect this curve, and one has to find the Riemann surface where one can follow continuously the curve. This can be done using the recent results of Ref. [92], by representing the polylogarithm as a sum of Hurwitz zeta functions (equation (57) in Ref. [92]), or as an infinite sum of square roots (equation (62) in Ref. [92]) with well-defined cuts. We were able to achieve this program numerically with an arbitrary precision, but when computing the polylogarithm $\text{Li}_{5/2}$ of Eq. (3.10) to determine $\psi_Q^{\text{ASEP}}(\mu)$, and inserting the result in the mapping (3.7) to obtain $\psi_K^{\text{SSEP}}(\mu)$, one does not obtain a real result. This implies that the limits $\varepsilon \to 0$ and $L \to \infty$ do not commute.

The question of computing the series (3.33) for $s > 0$ thus remains open, and would yield an interesting description of the hyperuniform phase beyond the diffusive regime described in Sec. 3.2. The aim would be to find the equivalent of the parametric representation of Eqs. (3.18)-(3.21) for $\psi_K^{\text{SSEP}}(s)$, but valid for $s > 0$.

In Sec. 3.7, we determine how geometrical properties of the SSEP in the hyperuniform phase are encoded in the structure factor, after having made explicit in Sec. 3.6 the relation to ground state of a spin chain.

## 3.6 Relation to the ground state of a spin chain

For the SSEP, the dominant eigenvector and eigenvalue of the evolution operator $M$ defined in Eq. (2.3) for the activity correspond to the ground state of the spin chain of $L$ sites

$$\hat{\mathbb{H}} = -M = -\frac{1}{2}e^s \sum_{1 \le k \le L} \left[ \sigma_k^x \sigma_{k+1}^x + \sigma_k^y \sigma_{k+1}^y + e^{-s}\sigma_k^z \sigma_{k+1}^z \right], \qquad (3.34)$$

with periodic boundary conditions $L + 1 \equiv 1$. To recast this chain in a common form, we perform every two sites (i.e. on all even or all odd sites) a rotation of angle $\pi$ of the spin operator triplet around the axis $z$, which transforms $\sigma^{x,y} \mapsto -\sigma^{x,y}$ and leaves $\sigma^z$ invariant. For the transformation to be well-defined we will suppose that $L$ is even. It is realized by a conjugation by $\otimes_{n=1}^{L/2} \sigma_{2n}^z$. We arrive at:

$$\mathbb{H} = J_0 \sum_{1 \le k \le L} \left[ \sigma_k^x \sigma_{k+1}^x + \sigma_k^y \sigma_{k+1}^y + \Delta \sigma_k^z \sigma_{k+1}^z \right] \qquad \text{with } J_0 = \frac{1}{2}e^s, \ \Delta = -e^{-s}, \qquad (3.35)$$

whose spectral properties are the same as those of $-M$. Due to the form of the transformation, correlation functions of $\sigma^z$ matrices will be identical for $-M$ and for $\mathbb{H}$. Since $J_0 > 0$, the Hamiltonian $\mathbb{H}$ describes a ferromagnetic XXZ spin chain, and, in the large-$L$ limit, the "ferromagnetic Heisenberg" point $\Delta = -1$ (i.e. $s = 0$) separates a ferromagnetic phase for $\Delta < -1$ (i.e. for the inactive phase $s < 0$) from an XY or 'Luttinger liquid' phase for $-1 < \Delta < 1$ (which contains the active phase $s > 0$). See for instance Ref. [93] for a review on 1D spin chains.

An important aspect of the statistical mechanics settings we are working in, is that the periodic exclusion processes we consider are isolated: their total number of particle $N$ is conserved. This implies that we are interested in the spectral elements of spin chains in a sector of fixed total magnetization, in contrast to the most standard condensed-matter viewpoint in which the ground state is found among states of any total magnetization. This is peculiarly

important in the vicinity of the $s = 0$ transition point, where, for $s < 0$, the ground state at fixed magnetization presents domain walls (a kink and an antikink, see [51] for the SSEP) while if one optimizes over the possible values of the total magnetization, one obtains two degenerate fully polarized states (all sites are empty or full).

## 3.7 Structure factor in the hyperuniform phase

The determination of the ground-state connected correlation function $\langle \sigma_0^z \sigma_n^z \rangle_c$ has been the subject of an extensive body of work. In the large $L$ limit, the so-called "massless phase" $-1 < \Delta < 1$ is well described by the Luttinger liquid theory [94–96] or by a conformal field theory [97–102]. Those allow one to write

$$\langle \sigma_0^z \sigma_n^z \rangle_c = \frac{c_1}{n^2} + (-1)^n c_2 \frac{\cos(\pi n \bar{m})}{n^\theta} + o(n^{-\theta}) \qquad \text{for } n \to \infty, \tag{3.36}$$

where $a$ and $b$ are constants and $\bar{m}$ is the average magnetization. Quantum inverse scattering methods [103, 104] and form factor approaches [105] have been used to assess this result rigorously and to obtain the rest $o(n^{-\theta})$ as a complete series in powers of $1/n^\theta$. See also the numerical studies of [106]. In the regime we are interested in ($s > 0$), one has $-1 < \Delta < 0$ and defining

$$\eta = \frac{1}{2} \arccos e^{-s} \underset{s \to 0}{\sim} \sqrt{\frac{s}{2}} \tag{3.37}$$

the exponent $\theta$ is given by

$$\theta = \frac{\pi}{2\eta} \underset{s \to 0}{\sim} \frac{\pi}{\sqrt{2s}}, \tag{3.38}$$

so that, as $s \to 0$ (and in fact for all $s > 0$), the $1/n^2$ term dominates the term in $1/n^\theta$ in the correlation function (3.36). The constant $c_1$ depends on the average magnetization and its value is known analytically only at half-filling (that is, for $N = L/2$) [107]:

$$c_1 = -\frac{1}{2\pi\eta} \underset{s \to 0}{\sim} -\frac{1}{\pi\sqrt{2s}}. \tag{3.39}$$

We emphasize that the result (3.36) is obtained by taking the limit $L \to \infty$ first, and then at large $n$. For a large but finite number of sites, one expects it is valid in the regime

$$1 \ll n \ll L. \tag{3.40}$$

The structure factor (which is the Fourier transform of the connected correlation function) inferred from (3.36) will also be valid only in a given range of Fourier modes: we now detail how its expected scaling is derived, keeping $L$ finite. For finite $L$ we define a discrete structure factor

$$\hat{S}_k = \sum_{-\frac{L}{2} \le n < \frac{L}{2}} e^{\frac{2i\pi k n}{L}} \langle \sigma_0^z \sigma_n^z \rangle_c \qquad \text{for integers } \frac{L}{2} \le k < \frac{L}{2}. \tag{3.41}$$

To go to a continuous structure factor we set

$$\mathfrak{q} = \frac{2\pi k}{L} \qquad \text{with } -\pi \le \mathfrak{q} < \pi \tag{3.42}$$

so that, defining $S(\mathfrak{q}) = \hat{S}_{k=\frac{L\mathfrak{q}}{2\pi}}$, one obtains in the large $L$ limit

$$S(\mathfrak{q}) = \sum_{n \in \mathbb{Z}} e^{i\mathfrak{q}n} \langle \sigma_0^z \sigma_n^z \rangle_c. \tag{3.43}$$

To use the result (3.36), we now split the sum in three contributions: one for $|n| \leq n_0$ which gives a constant (and $n_0$ is large enough for (3.36) to be valid); one for $n_0 < |n| < N_0$ where we can use the dominant contribution $\propto \frac{1}{n^2}$ of (3.36) (and $N_0$ is very small compared to the system size $L$); one for $|n| \geq N_0$ which gives a constant that goes to 0 as $N_0 \to \infty$ (as seen by a simple bound). This yields:

$$S(\mathfrak{q}) = \text{Constant} + c_1 \left[ \text{Li}_2(e^{i\mathfrak{q}}) + \text{Li}_2(e^{-i\mathfrak{q}}) \right], \tag{3.44}$$

where $\text{Li}_2$ is a polylogarithm function. This estimate is valid for large but finite $L$, in the regime

$$\frac{1}{L} \ll |\mathfrak{q}| < \pi. \tag{3.45}$$

To study the large-scale features, we now focus on the regime of values of $\mathfrak{q}$ which is insensitive to both the lattice spacing and the periodicity

$$\frac{1}{L} \ll |\mathfrak{q}| \ll \pi, \tag{3.46}$$

for which, expanding (3.44) for $\mathfrak{q} \to 0$ (but still respecting (3.46)), one finds

$$S(\mathfrak{q}) = \text{Constant}' - c_1 \pi |\mathfrak{q}| + O(\mathfrak{q}^2). \tag{3.47}$$

The constant can be inferred from the exact relation $\hat{S}_{k=0} = 0$ that comes from the strict conservation of magnetization. Using (3.39), this yields

$$S(\mathfrak{q}) = \frac{|\mathfrak{q}|}{\sqrt{2s}} + O(\mathfrak{q}^2) \qquad \text{for } s \to 0 \text{ and at half-filling}, \tag{3.48}$$

and again with the finite-$L$ condition (3.46) on $\mathfrak{q}$.

Strikingly, the EW diffusive computation of MFT gives the same scaling for the structure factor [53, 108], including the same numerical prefactor[4]. However, the MFT computation is performed in the regime where $s = \lambda/L^2$ and $L$ is large, with then $\lambda$ sent to infinity. The fact that the two asymptotic regimes $\lambda \to \infty$ of the diffusive scaling and $s \to 0$ of the result (3.48) match, indicate that the scaling $S(\mathfrak{q}) = \frac{|\mathfrak{q}|}{\sqrt{2s}}$ of the hyperuniform phase is very robust.

To get a more physical intuition of the hyperuniform regime, another standard characterization of anomalous fluctuations is to look at the variance $\sigma_n^2(\ell)$ of the number $n$ of particles enclosed in a virtual box of linear size $\ell$. The length $\ell$ is assumed to be large with respect to microscopic sizes, like lattice spacing or particle size, but small with respect to system size. Considering a generic $d$-dimensional system, normal fluctuations correspond to $\sigma_n^2(\ell) \sim \ell^d$ (we discard here numerical prefactors). This is what happens in the absence of long-range correlations in the system. A different scaling of $\sigma_n^2(\ell)$ means that fluctuations are anomalous. Fluctuations growing faster with system size, as $\sigma_n^2(\ell) \sim \ell^\zeta$ with $\zeta > d$ are found for instance in the ordered phases of systems of active particles with alignment interactions [109], where they are called giant number fluctuations (in this context, one often plots $\sigma_n^2$ as a function of $\langle n \rangle$, parametrized by $\ell$). The opposite situation, namely $\sigma_n^2(\ell) \sim \ell^\zeta$ with an exponent $\zeta < d$, corresponds to hyperuniformity. An extreme example of hyperuniform arrangement is the perfect crystal, for which $\zeta = 0$. It can be shown that the exponent $\zeta$ is related to the small-wavenumber behavior of the structure factor. If $S(\mathfrak{q}) \sim |\mathfrak{q}|^\eta$ for $|\mathfrak{q}| \to 0$ (discarding again prefactors), then $\zeta = d - \eta$ [110, 111]. When $\eta = 1$, logarithmic corrections are present, meaning that $\sigma_n^2(\ell) \sim \ell^{d-1} \log \ell$ [110, 111]. In the present one-dimensional ASEP

---

[4]Note that in the MFT the structure factor is naturally computed for the density: $S^{\text{occ.}}(\mathfrak{q}) = \int dx\, e^{i\mathfrak{q}x} \langle \rho(x)\rho(0) \rangle_c$. Since the macroscopic density corresponds to the microscopic occupation number $\frac{1}{2}(\sigma^z + 1)$, the spin-spin structure factor (3.48) and the occupation structure factor are related by $S^{\text{occ.}}(\mathfrak{q}) = \frac{1}{4}S(\mathfrak{q})$.

model conditioned to have a large activity, we thus find $\sigma_n^2(\ell) \sim \log \ell$, so that fluctuations of particle number grow only logarithmically with the box size $\ell$. Hence particle configurations are almost as ordered as in a crystal.

Last, the MFT computation for the SSEP predicts that for any average density $\bar{\rho}$, the spin-spin structure factor is

$$S(\mathfrak{q}) \sim |\mathfrak{q}| \sqrt{\frac{2\bar{\rho}(1-\bar{\rho})}{s}} \quad \text{with} \ \ s = \frac{\lambda}{L^2} \ \text{and} \ \lambda \to \infty \,. \tag{3.49}$$

If the matching observed at half-filling between the MFT computation at $\lambda \to \infty$ and the Luttinger-liquid result (3.36) at $s \to 0$ holds, we obtain the conjecture that for $s \to 0^+$ (i.e. for $\Delta \to -1^+$) the constant $c_1$ in (3.36) behaves as:

$$c_1 \underset{s \to 0^+}{\sim} -\frac{1}{\pi} \sqrt{\frac{2\bar{\rho}(1-\bar{\rho})}{s}} \,, \tag{3.50}$$

which generalizes the half-filling result (3.39). In terms of the average magnetization $\bar{m} = 2\bar{\rho}-1$ (with $-1 < \bar{m} < 1$) and the parameter $\Delta$ of the XXZ chain (3.35), this conjecture is reexpressed as:

$$c_1 \underset{\Delta \to -1^+}{\sim} -\frac{1}{\pi} \sqrt{\frac{1-\bar{m}^2}{2(1+\Delta)}} \,, \tag{3.51}$$

with $c_1$ the constant in the expansion (3.36) of the correlation function.

To summarize, we have shown that, for closed systems, the fluctuations of activity are characterized at large activity by successively a diffusive EW regime and a KPZ regime (on one side of the transition). We have discussed in Sec. 3.5 why studying the other side of the transition beyond the diffusive regime proves to be difficult. We have shown that (at least at half-filling) however, the structure factor is a very robust observable that behaves in the same way in these two regimes (at least at scales which are insensitive to the lattice spacing or to the system periodicity). We now analyze how the mapping between joint SCGFs is implemented for open systems and we discuss its consequences on an example DPT.

# 4 Systems in contact with reservoirs

Having studied above the scaled cumulant generating function of activity and current for the case of ASEPs with periodic boundary conditions, we now explore the possibility to extend the mapping to open ASEPs connected to boundary reservoirs. We then discuss specific cases, and connect this microscopic mapping to the MFT description.

## 4.1 General mapping between ASEPs with open boundaries

The open ASEP is described by the bulk jump rates $p$ and $q$ as in the periodic case, as well as by transition rates $\alpha$, $\beta$, $\gamma$ and $\delta$ with the two reservoirs. More specifically, a particle is injected from the left reservoir to the leftmost site $i = 1$ with rate $\alpha$ if this site is empty. A particle sitting on site $i = 1$ jumps to the reservoir with rate $\gamma$. Similarly, the right reservoir injects a particle onto the rightmost site $i = L$ with rate $\beta$, and particles jump from site $i = L$ to the right reservoir with rate $\delta$.

We consider two open ASEPs conditioned to average values of activity and current by parameters $s_j$ and $\mu_j$ ($j = 1, 2$). The mapping between the two ASEPs is performed in the same way as for the periodic case, except that boundary reservoirs now need to be taken

into account. We denote as $W$ and $M$ the deformed Markov matrices of the first and second ASEPs respectively. The corresponding local deformed Markov matrices describing the bulk dynamics are called $w_{k,k+1}(s_1, \mu_1)$ and $m_{k,k+1}(s_2, \mu_2)$. The exchange of particles with boundary reservoirs are described by boundary deformed Markov matrices $B_1$ (left reservoir) and $\bar{B}_L$ (right reservoir) for system 1, and respectively boundary matrices $B_1'$ and $\bar{B}_L'$ for system 2. The deformed Markov matrices read

$$W = B_1 + \bar{B}_L + \sum_{k=1}^{L-1} w_{k,k+1}(s_1, \mu_1) \quad \text{with} \quad B = \begin{pmatrix} -\alpha_1 & \gamma_1 e^{s_1 - \mu_1} \\ \alpha_1 e^{s_1 + \mu_1} & -\gamma_1 \end{pmatrix}, \bar{B} = \begin{pmatrix} -\delta_1 & \beta_1 e^{s_1 + \mu_1} \\ \delta_1 e^{s_1 - \mu_1} & -\beta_1 \end{pmatrix} \tag{4.1}$$

$$M = B_1' + \bar{B}_L' + \sum_{k=1}^{L-1} m_{k,k+1}(s_2, \mu_2) \quad \text{with} \quad B' = \begin{pmatrix} -\alpha_2 & \gamma_2 e^{s_2 - \mu_2} \\ \alpha_2 e^{s_2 + \mu_2} & -\gamma_2 \end{pmatrix}, \bar{B}' = \begin{pmatrix} -\delta_2 & \beta_2 e^{s_2 + \mu_2} \\ \delta_2 e^{s_2 - \mu_2} & -\beta_2 \end{pmatrix}. \tag{4.2}$$

Defining the matrix $U = \otimes_{k=1}^L V_k$ with $V_k = \begin{pmatrix} 1 & 0 \\ 0 & u\Lambda^{k-1} \end{pmatrix}$, we look for a relation between the matrices $M$ and $W$ of the form

$$M = \frac{p_2 + q_2}{p_1 + q_1} U W U^{-1} + (\epsilon + \tau)\mathbb{I}, \tag{4.3}$$

which is a similarity transformation up to a global numerical factor and a shift by a matrix proportional to identity. The parameters $u$, $\Lambda$, $\epsilon$ and $\tau$ appearing in Eq. (4.3) are unknown at this stage and will be determined below. With the addition of telescopic terms, Eq. (4.3) locally reads

$$m_{k,k+1}(s_2, \mu_2) = \frac{p_2 + q_2}{p_1 + q_1} V_k V_{k+1} w_{k,k+1}(s_1, \mu_1) V_k^{-1} V_{k+1}^{-1} - \frac{p_1 q_2 - p_2 q_1}{2(p_1 + q_1)}\left(\sigma_k^z - \sigma_{k+1}^z\right), \tag{4.4}$$

$$B' = \frac{p_2 + q_2}{p_1 + q_1} V_1 B V_1^{-1} + \frac{p_1 q_2 - p_2 q_1}{2(p_1 + q_1)}\sigma^z + \epsilon\,\mathbb{I}, \tag{4.5}$$

$$\bar{B}' = \frac{p_2 + q_2}{p_1 + q_1} V_L \bar{B} V_L^{-1} - \frac{p_1 q_2 - p_2 q_1}{2(p_1 + q_1)}\sigma^z + \tau\,\mathbb{I}. \tag{4.6}$$

These relations hold provided the following constraints are satisfied

$$\mu_1 - \mathcal{E}_1 = \mu_2 - \mathcal{E}_2 - \log(\Lambda) \quad \text{with} \quad \mathcal{E}_j = \log\sqrt{\frac{q_j}{p_j}}, \; j = 1, 2 \tag{4.7}$$

$$s_1 - \log\cosh(\mathcal{E}_1) = s_2 - \log\cosh(\mathcal{E}_2)$$

$$\frac{\alpha_1 - \gamma_1}{p_1 + q_1} - \frac{\alpha_2 - \gamma_2}{p_2 + q_2} = \frac{p_1 q_2 - p_2 q_1}{(p_1 + q_1)(p_2 + q_2)} \tag{4.8}$$

$$\frac{\beta_1 - \delta_1}{p_1 + q_1} - \frac{\beta_2 - \delta_2}{p_2 + q_2} = \frac{p_1 q_2 - p_2 q_1}{(p_1 + q_1)(p_2 + q_2)} \tag{4.9}$$

$$\frac{\alpha_2 \gamma_2}{p_2 q_2} = \frac{\alpha_1 \gamma_1}{p_1 q_1}, \quad \frac{\beta_2 \delta_2}{p_2 q_2} = \frac{\beta_1 \delta_1}{p_1 q_1} \tag{4.10}$$

$$\epsilon = \frac{p_2 + q_2}{p_1 + q_1} \frac{\alpha_1 + \gamma_1}{2} - \frac{\alpha_2 + \gamma_2}{2} \tag{4.11}$$

$$\tau = \frac{p_2 + q_2}{p_1 + q_1} \frac{\beta_1 + \delta_1}{2} - \frac{\beta_2 + \delta_2}{2}. \tag{4.12}$$

The parameters $u$ and $\Lambda$ are given by

$$u = \frac{p_1 \alpha_2}{p_2 \alpha_1}\left(\frac{p_2 q_1 \alpha_1 \delta_2}{p_1 q_2 \alpha_2 \delta_1}\right)^{\frac{1}{L+1}}, \quad \Lambda = \left(\frac{p_2 q_1 \alpha_1 \delta_2}{p_1 q_2 \alpha_2 \delta_1}\right)^{L+1}. \tag{4.13}$$

Then, when the above relations are fulfilled, we get the following connection between scaled cumulant generating functions:

$$\frac{1}{p_1 + q_1}\Big(\psi(\mu_1, s_1 | p_1, q_1, \alpha_1, \gamma_1, \beta_1, \delta_1) - (\alpha_1 + \delta_1)\Big) =$$
$$= \frac{1}{p_2 + q_2}\Big(\psi\big(\mu_1 - E_1 + E_2, s_1 - \log\cosh(\mathcal{E}_1) + \log\cosh(\mathcal{E}_2)\big| p_2, q_2, \alpha_2, \gamma_2, \beta_2, \delta_2\big) - (\alpha_2 + \delta_2)\Big),$$
(4.14)

where we have introduced

$$E_j = \frac{L-1}{L+1}\mathcal{E}_j + \frac{1}{2(L+1)}\log\left(\frac{\gamma_j \delta_j}{\alpha_j \beta_j}\right), \qquad j = 1, 2. \tag{4.15}$$

We discuss in Appendix B another parametrisation of the mapping (4.14) which involves the densities of the reservoirs at boundaries.

**Gallavotti-Cohen symmetry point**

Using (4.15), the first relation of (4.7) can be rewritten in a more symmetric way:

$$\mu_1 - E_1 = \mu_2 - E_2. \tag{4.16}$$

The GC symmetry of the ASEP with open boundary conditions is then expressed as [112]

$$\psi(\mu, s | p, q, \alpha, \gamma, \beta, \delta) = \psi(2E - \mu, s | p, q, \alpha, \gamma, \beta, \delta) \tag{4.17}$$

meaning that the function $\psi$ is symmetric around $E$ in terms of its argument $\mu$. The relation (4.16) implies that the GC symmetry of the target model coincides with the GC symmetry of the source model.

## 4.2 Simpler mappings in specific cases

The above general mapping (4.14) between the SCGFs of two open ASEPs can be simplified in some specific cases. We provide in this section a few examples of simpler mappings that may be of physical relevance.

**Case of reservoirs with same densities.**

To get an example similar to the periodic case, we study the case where reservoirs have the same density. For an ASEP model with parameters $(p, q, \alpha, \beta, \gamma, \delta)$, the reservoir densities take the form [113, 114]

$$\rho_a = \frac{1}{1+a}, \quad a = \frac{1}{2\alpha}\Big(p - q - \alpha + \gamma + \sqrt{(p - q - \alpha + \gamma)^2 + 4\alpha\gamma}\Big)$$
$$\rho_b = \frac{1}{1 + 1/b}, \quad b = \frac{1}{2\beta}\Big(p - q + \delta - \beta + \sqrt{(p - q + \delta - \beta)^2 + 4\delta\beta}\Big). \tag{4.18}$$

The densities $\rho_a$ and $\rho_b$ are equal if, on only if, the following relation holds:

$$(p - q)(\alpha + \delta)(\gamma + \beta) = (\alpha + \beta + \gamma + \delta)(\alpha\beta - \gamma\delta). \tag{4.19}$$

In that case, the density reads

$$\rho = \frac{\alpha + \delta}{\alpha + \beta + \gamma + \delta}. \tag{4.20}$$

A particular solution to (4.19) is given by $\alpha = \gamma + \frac{p-q}{2}$ and $\beta = \delta + \frac{p-q}{2}$. Remark that this solution corresponds (in fact is equivalent) to $\alpha + \delta = \beta + \gamma$. In that case, one gets $\rho_a = \rho_b = \frac{1}{2}$.

Now, we start with a model such that $\alpha_1 - \gamma_1 = \frac{p_1 - q_1}{2}$ and $\beta_1 - \delta_1 = \frac{p_1 - q_1}{2}$, so that $\rho_a^{(1)} = \rho_b^{(1)} = \frac{1}{2}$. Then, using relations (4.8)-(4.9) we get

$$\alpha_2 = \gamma_2 + \frac{p_2 - q_2}{2} \quad \text{and} \quad \beta_2 = \delta_2 + \frac{p_2 - q_2}{2}, \tag{4.21}$$

so that $\rho_a^{(2)} = \rho_b^{(2)} = \frac{1}{2}$.

**Connecting SSEP with current to ASEP with activity.** We impose $p_1 = q_1 = 1$ and $s_1 = 0 = \mu_2$. In that case, using formulas (4.7) and (4.10), we deduce

$$\mu_1 = \left(\frac{1}{2} - \frac{1}{L}\right)\log\left(\frac{q_2}{p_2}\right) - \frac{1}{L+1}\log\left(\frac{\alpha_2 \delta_1}{\alpha_1 \delta_2}\right) \quad ; \quad s_2 = \log\frac{1}{2}\left(\sqrt{\frac{q_2}{p_2}} + \sqrt{\frac{p_2}{q_2}}\right). \tag{4.22}$$

The boundary terms do not contribute at leading order in $L$.

**Models with same bulk parameters.** Note also that if we set $p_1 = p_2$ and $q_1 = q_2$ (connection of two models with same bulk parameters), the equations (4.8)-(4.10) simplify to

$$\begin{cases} \alpha_1 - \gamma_1 = \alpha_2 - \gamma_2 \\ \alpha_1 \gamma_1 = \alpha_2 \gamma_2 \end{cases} \quad \text{and} \quad \begin{cases} \beta_1 - \delta_1 = \beta_2 - \delta_2 \\ \beta_1 \delta_1 = \beta_2 \delta_2 \end{cases}. \tag{4.23}$$

Since all parameters must be positive, we deduce that the boundary parameters stay the same. We also obtain through relations (4.7) that $s_1 = s_2$ and $\mu_1 = \mu_2$. In other words, we get the same model. The transformations used in this paper, to be non-trivial, must connect models with different values of $(p, q)$.

## 4.3 Diffusive scaling: MFT and WASEP

We now briefly discuss the above mapping between open systems in the diffusive scaling limit where the asymmetry of the bulk rates scales as $1/L$ and the macroscopic time scales as $1/L^2$. We first reinterpret the mapping in the MFT context, and then apply the mapping to the case of two WASEP, unraveling the existence of a phase transition in the WASEP conditioned by the activity.

**Relation with MFT in the open case**

In the WASEP limit, defined by

$$p_j = \exp\left(\frac{v_j}{L}\right) \quad \text{and} \quad q_j = \exp\left(-\frac{v_j}{L}\right), \qquad j = 1, 2 \tag{4.24}$$

one has

$$\Lambda = 1 - \frac{v_1 - v_2}{L^2}\left(2 + \frac{1}{\beta_1 + \delta_1} + \frac{1}{\alpha_1 + \gamma_1}\right) + o\left(\frac{1}{L^2}\right)$$

$$\text{and} \quad u = 1 + \frac{v_1 - v_2}{L}\left(1 + \frac{1}{\alpha_1 + \gamma_1}\right) + o\left(\frac{1}{L}\right),$$

$$x = 1 + \frac{v_1 - v_2}{L}\left(1 + \frac{1}{\alpha_1 + \gamma_1}\right) + o\left(\frac{1}{L}\right)$$

$$\text{and} \quad y = 1 + \frac{v_1 - v_2}{L}\left(1 + \frac{1}{\beta_1 + \delta_1}\right) + o\left(\frac{1}{L}\right). \tag{4.25}$$

Then, the relations for $\mu_1$, $\mu_2$, $s_1$ and $s_2$ are the same as in the periodic case (section 2.4) up to irrelevant orders in $1/L$. The boundary parameters $\alpha_2$, $\beta_2$, $\gamma_2$ and $\delta_2$ become

$$\alpha_2 = \alpha_1 + \frac{\nu_1 - \nu_2}{L}\rho_a + o\left(\frac{1}{L}\right), \qquad \gamma_2 = \gamma_1 - \frac{\nu_1 - \nu_2}{L}(1 - \rho_a) + o\left(\frac{1}{L}\right), \quad \rho_a = \frac{\alpha_1}{\alpha_1 + \gamma_1},$$

$$\beta_2 = \beta_1 + \frac{\nu_1 - \nu_2}{L}(1 - \rho_b) + o\left(\frac{1}{L}\right), \quad \delta_2 = \delta_1 - \frac{\nu_1 - \nu_2}{L}\rho_b + o\left(\frac{1}{L}\right), \qquad \rho_b = \frac{\delta_1}{\beta_1 + \delta_1}.$$
$$(4.26)$$

As in the periodic case, the relation (4.14) can then be interpreted in the WASEP limit. Starting again from the relation (2.19), but taking into account the boundary values $\rho(0, t) = \rho_a$ and $\rho(1, t) = \rho_b$, we obtain:

$$\mathcal{A}_2(\rho, j) = \mathcal{A}_1(\rho, j) + (\nu_1 - \nu_2)\int_0^T dt \int_0^1 dx\, j + (\nu_2^2 - \nu_1^2)\int_0^T dt \int_0^1 dx\, \rho(1 - \rho)$$
$$- T(\nu_2 - \nu_1)(\rho_a - \rho_b). \qquad (4.27)$$

To have a full correspondence, one has to remark that in the WASEP limit, $\epsilon + \tau = \frac{1}{L}(\nu_2 - \nu_1)(\rho_a - \rho_b)$. Indeed, the action on the MFT side occurs with a factor $L$, that matches the factor $\frac{1}{L}$ in $\epsilon + \tau$ once the diffusive rescaling $T = t/L^2$ is made.

In the case of reservoirs with same densities, one needs to go to the next order in the $1/L$ expansion for $\epsilon + \tau$, and we obtain

$$\epsilon + \tau = \frac{(\nu_1 - \nu_2)(\alpha_1 + \beta_1)}{L^2}\frac{4\alpha_1\beta_1(\nu_1 + \nu_2) + \nu_1 - \nu_2}{8\alpha_1\beta_1}, \qquad (4.28)$$

which corresponds to $\epsilon + \tau = 0$ for the MFT.

**Connecting WASEPs conditioned by activity or current at densities $\frac{1}{2}$**

For a WASEP model with bulk asymmetry parameter $\nu$ (defined by $p/q = e^{2\nu/L}$) and reservoir densities $\rho_a = \rho_b = \frac{1}{2}$, it has been shown [54] that for current large deviations with $\mu = \frac{r}{L}$ (with $r$ fixed and $L \to \infty$), and for large enough asymmetry amplitude $|\nu| > \nu_c$ with $\nu_c = \pi$, there are particle-hole symmetry breaking transitions at $r = r_c^\pm = -\nu \pm \sqrt{\nu^2 - \pi^2}$. For $r < r_c^-$ or $r > r_c^+$, the particle-hole symmetry is broken and the optimal density profile is no longer flat, but remains time-independent (meaning that the additivity property remains valid) [54]. Note that the criterion for symmetry breaking can be reformulated in a concise way as $|r + \nu| > \sqrt{\nu^2 - \pi^2}$, which will prove useful in the following.

We now exploit the mapping described above to deduce the existence of a dynamical phase transition in the large deviations of activity in the WASEP. We set $\alpha_j = \gamma_j + \frac{\nu_j}{L}$ and $\beta_j = \delta_j + \frac{\nu_j}{L}$ ($j = 1, 2$), where $\nu_j$ is the bulk bias parameter defined in Eq. (4.24). As already discussed, see (4.21), this ensures that both models have densities $\rho_a = \rho_b = \frac{1}{2}$. We consider that model 1 (source model) is conditioned by activity ($\mu_1 = 0$) and that model 2 (target model) is conditioned by the current ($s_2 = 0$). At leading order in $\frac{1}{L}$, we get from Eq. (4.7) that

$$s_1 = \frac{\nu_1^2 - \nu_2^2}{2L^2} \quad \text{and} \quad \mu_2 = \frac{\nu_1 - \nu_2}{L}. \qquad (4.29)$$

Writing $s_1 = \frac{\lambda}{L^2}$ and $\mu_2 = \frac{r}{L}$, we get $\lambda = \frac{1}{2}(\nu_1^2 - \nu_2^2)$ and $r = \nu_1 - \nu_2$. We fix $\nu_1$ and look for a possible symmetry breaking transition in model 1 by varying $\lambda$, using the mapping to model 2 for which we know that a symmetry breaking transition occurs in a given parameter range. From the mapping we get $|\nu_2| = \sqrt{\nu_1^2 - 2\lambda}$, which imposes a first restriction $\lambda \leq \frac{\nu_1^2}{2}$ to the range of $\lambda$ that can be explored through the mapping. The criterion for particle-hole symmetry

breaking in system 2 reads $|r + \nu_2| > \sqrt{\nu_2^2 - \pi^2}$, provided that $|\nu_2| > \pi$. Using $r = \nu_1 - \nu_2$ and the above expression of $|\nu_2|$ in terms of $\nu_1$ and $\lambda$, these two conditions respectively turn into $\lambda > -\frac{\pi^2}{2}$ and $\lambda < \frac{\nu_1^2 - \pi^2}{2}$. We thus conclude that particle-hole symmetry is broken in the WASEP conditioned by activity (and with boundary reservoirs at densities $\rho_a = \rho_b = \frac{1}{2}$) in the range

$$-\frac{\pi^2}{2} < \lambda < \frac{\nu_1^2 - \pi^2}{2}. \tag{4.30}$$

In this parameter range, optimal density profiles are not flat. The corresponding density profiles can be deduced from those obtained in [54] in the WASEP conditioned by the current. In contrast, for $\lambda \leq -\frac{\pi^2}{2}$ and for $\frac{\nu_1^2 - \pi^2}{2} \leq \lambda \leq \frac{\nu_1^2}{2}$, the particle-hole symmetry holds and optimal profiles are flat. Note that the range $\lambda > \frac{\nu_1^2}{2}$ is not accessible to the mapping (as long as one keeps real values of the parameters), so that we cannot conclude whether particle-hole symmetry is broken or not in this range.

One sees from Eq. (4.30) that the extension of the range of $\lambda$ over which symmetry breaking occurs vanishes for $\nu_1 = 0$. Hence there is no symmetry breaking in the SSEP, but one might expect to see in this model some kind of critical behavior around $\lambda = -\frac{\pi^2}{2}$, with possibly the onset of power-law correlations.

# 5 Conclusion and outlook

We have shown that a generic mapping between the joint SCGF of the current and activity between two generic ASEP models can be used to uncover a surprising regime where KPZ scalings govern the fluctuations of activity in the SSEP. The occurrence of such KPZ scaling could be interestingly compared to those recently uncovered by Prosen and coworkers for $SO(3)$ classical spin dynamics [115] or for the quantum Heisenberg magnet [116].

Thermodynamic Bethe Ansatz and its extensions allows one to obtain corrections to the EW scaling (3.5) at large $\lambda$, as done in Ref. [58]: it could be instructive to see how the expansion matches the next orders of the small-$s$ expansion (3.22)-(3.23) in the KPZ regime. Related to this, the existence of two different scalings with $L$ (EW and KPZ) on each side of the transition could be interesting to describe how the coordinate Bethe Ansatz behaves as one crosses over from the EW scaling regime ($s = \lambda/L^2$ with fixed $\lambda$ and large $L$), studied in Ref. [47], to the KPZ scaling regime (finite $s$ and large $L$). The question remains open to understand how the series (3.33) can be resummed in order to understand how the SCGF $\psi_K^{\text{SSEP}}(s)$ behaves in the hyperuniform regime $s > 0$ for $s = O(L^0)$ (i.e. beyond the diffusive regime). Since the singularities occur when $\frac{q}{p}$ are unit roots, it would be interesting to see if the results of Refs. [117–120] could be used in some way.

Also, the mapping we have used could be related to other ones that have been found for the current LDFs [19, 20] or for the steady-state large deviations [121, 122]. Last, studies of the gap in the deformed evolution operator (which is in principle possible within the Bethe Ansatz approach) could shed light on the finite-time scalings close to a DPT, in the spirit of previous works in diffusive [123] and super-diffusive [124] models, or in so-called large-$N$ models [55, 125]. For instance, the gap of $W$ for $\mu = s = 0$ has been obtained [73, 126, 127] for the TASEP and the ASEP in the absence of Legendre parameter conjugated to the current or the activity, and has been found to scale as $1/L^{3/2}$ at large $L$. It would be interesting to extend their computation to the case of the deformed Markov matrices we have considered. For instance, if through the mapping between $M$ and $W$, the gap of $M$ scales as $1/L^{3/2}$ for large enough $|s|$, it would imply that the dynamical exponent of the large deviations of activity in the SSEP is 3/2 for deviations beyond the diffusive regime (where the dynamical exponent is 2).

# Acknowledgments

The authors acknowledge financial support from the grant IDEX-IRS 'PHEMIN' of the Université Grenoble Alpes. VL acknowledges support by the ERC Starting Grant No. 680275 MALIG, the ANR-18-CE30-0028-01 Grant LABS and the ANR-15-CE40-0020-03 Grant LSD.

# Appendices

## A Imaginary $\mu$

We recall (see Sec. 2.2) that the matrix $W = \sum_{k=1}^{L} w_{k,k+1}$ with

$$w = \begin{pmatrix} 0 & 0 & 0 & 0 \\ 0 & -q & pe^{\mu} & 0 \\ 0 & qe^{-\mu} & -p & 0 \\ 0 & 0 & 0 & 0 \end{pmatrix} \tag{A.1}$$

and periodic boundary conditions $L + 1 \equiv 1$ describes, for $\mu \in \mathbb{R}$ and $p > 0$, $q > 0$ the fluctuations of the current $Q$ of an ASEP of jump parameters $p$ and $q$, with $\mu$ being the Lagrange multiplier associated with $Q$. At fixed total number of particles $N$, the SCGF for the distribution of $Q$ is the eigenvalue $\psi_Q^{\text{ASEP}}(\mu)$ of $W$ which possesses the largest real part (see e.g. [1]). From the Perron–Frobenius theorem (which applies for $\mu \in \mathbb{R}$), it is unique and isolated. This means that $\psi_Q^{\text{ASEP}}(\mu)$ is an isolated zero of multiplicity 1 of the characteristic polynomial of $W$. Since the coefficients of this polynomial are analytic functions of $\mu \in \mathbb{C}$, one has (see e.g. [128]) that $\psi_Q^{\text{ASEP}}(\mu)$ is analytic as a function of the complex variable $\mu$, in the vicinity of every $\mu \in \mathbb{R}$. Besides, $\psi_Q^{\text{ASEP}}(\mu) \to 0$ as $\mu \to 0$ by conservation of probability. All this implies that one can identify $\psi_Q^{\text{ASEP}}(\mu)$ for $\mu \in \mathbb{R}$ as the unique branch of the spectrum of $W$ that continuously goes to 0 as $\mu \to 0$, and that $\psi_Q^{\text{ASEP}}(\mu)$ can be analytically continued as function of $\mu$ on an open subset of $\mathbb{C}$ that contains $\mathbb{R}$.

In the Bethe Ansatz computation of Prolhac (Ref. [81]), the SCGF $\psi_Q^{\text{ASEP}}(\mu)$ is precisely identified using the criterion that it is the isolated eigenvalue of $W$ that goes to 0 as $\mu \to 0$ (see Eq. (33) of [81]). Prolhac also gives an expression of the derivatives $\partial \psi_Q^{\text{ASEP}}(\mu)/\partial \mu^k$ in $\mu = 0$ at any order $k \geq 0$, and for any finite size $L$ and finite occupation $N$. The corresponding Taylor series is valid in a vicinity of 0 not only for $\mu \in \mathbb{R}$, but also in a vicinity of 0 for $\mu \in \mathbb{C}$, thanks to the analyticity property we mentioned above. This definition of $\psi_Q^{\text{ASEP}}(\mu)$ for small $|\mu|$ with $\mu \in \mathbb{C}$ is used in Sec. 3.3 when mapping the fluctuations of $Q$ in the ASEP to those of $K$ in the SSEP.

We have thus seen that the function $\psi_Q^{\text{ASEP}}(\mu)$ can be expanded as an entire function of the complex variable $\mu$ in a vicinity of $\mu = 0$. We now show that $\psi_Q^{\text{ASEP}}(\mu)$ can be analytically continued on the imaginary axis for any $\mu = i\mu'$ with $\mu' \in ]-\frac{\pi}{2}, \frac{\pi}{2}[$, for the parameters $p, q$ of the mapping of Eq. (2.9), i.e. for $\cos \mu' = e^{-s} \in ]0, 1]$ and $q/p = e^{2i\mu'}$. As seen from Eqs. (2.7) and (2.9), we have $\frac{1}{2}(p + q) = p\, e^{i\mu'} \cos \mu'$ and $W$ is thus the sum of local operators

$$p\, e^{i\mu'} \begin{pmatrix} 0 & 0 & 0 & 0 \\ 0 & -\cos \mu' & 1 & 0 \\ 0 & 1 & -\cos \mu' & 0 \\ 0 & 0 & 0 & 0 \end{pmatrix}. \tag{A.2}$$

Without the prefactor $p\, e^{i\mu'}$, the local operator gives a matrix that satisfies the conditions of the Perron–Frobenius theorem (we recall that $\cos \mu' = e^{-s} \in ]0, 1]$) at fixed total occupation

$N$ ; its maximal eigenvalue is isolated for all $\mu' \in\, ]-\frac{\pi}{2}, \frac{\pi}{2}[$, and as we discussed above, can be analytically continued as a function of the complex variable $\mu'$ in an open subset of $\mathbb{C}$ containing $]-\frac{\pi}{2}, \frac{\pi}{2}[$. To conclude, the same holds for $\psi_Q^{\mathrm{ASEP}}(\mu)$ with $\mu \in i\, ]-\frac{\pi}{2}, \frac{\pi}{2}[$, for the parameters $p, q$ of the mapping of Eq. (2.9).

To summarize, the mapping that we used in Sec. 2.2 between the LDF $\psi_K^{\mathrm{SSEP}}(s)$ of $K$ in the SSEP and the LDF $\psi_Q^{\mathrm{ASEP}}(\mu)$ of $Q$ in the ASEP can be used transparently both for $s \leq 0$ (which corresponds to real values of $\mu$, see Eq. (2.6)) and for $s > 0$ (corresponding to imaginary values of $\mu$, see Eq. (2.9)), when employing the results of Prolhac in Ref. [81] for the expression of $\psi_Q^{\mathrm{ASEP}}(\mu)$. Note that, contrarily to Kim and Lee [88], Prolhac works in finite system size and then takes the hydrodynamic limit $L, N \to \infty$ with fixed $\rho = N/L$. This means that the large-$L$ asymptotics for $\psi_Q^{\mathrm{ASEP}}(\mu)$, because they are taken as a large-$L$ limit of an analytic function of $\mu$, can also be used for $\mu$ in the complex domain of analyticity of $\psi_Q^{\mathrm{ASEP}}(\mu)$.

# B  Expression of the boundary parameters in term of the reservoir densities

In this Appendix, we discuss an alternative parameterization of the scaled cumulant generating function of current and activity in the ASEP in terms of possibly more physically meaningful parameters, like the asymmetry of the bulk drive, and the densities of the reservoir.

For any ASEP model with bulk parameters $(p, q)$ and boundary parameters $(\alpha, \beta, \gamma, \delta)$, we introduce the reservoir densities

$$\rho_a = \frac{2\alpha}{2\alpha + \mathfrak{a}} \quad \text{with} \quad \mathfrak{a} = p - q - \alpha + \gamma + \sqrt{(p - q - \alpha + \gamma)^2 + 4\alpha\gamma} \tag{B.1}$$

$$\rho_b = \frac{\mathfrak{b}}{2\beta + \mathfrak{b}} \quad \text{with} \quad \mathfrak{b} = p - q + \delta - \beta + \sqrt{(p - q + \delta - \beta)^2 + 4\delta\beta} \tag{B.2}$$

together with the following parameters $\sigma_a$ and $\sigma_b$ that compare the asymmetry of the reservoir transition rates with that of the bulk dynamics:

$$\sigma_a = \frac{\alpha - \gamma - \frac{1}{2}(p - q)}{p + q} \quad \text{and} \quad \sigma_b = \frac{\beta - \delta - \frac{1}{2}(p - q)}{p + q}. \tag{B.3}$$

The boundary parameters of the model can be reconstructed from the reservoir densities and currents:

$$\frac{\alpha}{p + q} = \frac{\rho_a}{2\rho_a - 1}\left(\sigma_a + \frac{p - q}{p + q}\left(\rho_a - \frac{1}{2}\right)\right) \quad \text{and} \quad \frac{\gamma}{p + q} = \frac{1 - \rho_a}{2\rho_a - 1}\left(\sigma_a - \frac{p - q}{p + q}\left(\rho_a - \frac{1}{2}\right)\right),$$

$$\frac{\beta}{p + q} = \frac{\rho_b}{2\rho_b - 1}\left(\sigma_b + \frac{p - q}{p + q}\left(\rho_b - \frac{1}{2}\right)\right) \quad \text{and} \quad \frac{\delta}{p + q} = \frac{1 - \rho_b}{2\rho_b - 1}\left(\sigma_b - + \frac{p - q}{p + q}\left(\rho_b - \frac{1}{2}\right)\right),$$

$$\tag{B.4}$$

where we have renormalized the boundary parameters by $p + q$ because of the connection (4.3), which can be expressed as

$$\frac{1}{p_2 + q_2}\left(M + \frac{1}{2}(\alpha_2 + \beta_2 + \gamma_2 + \delta_2)\mathbb{I}\right) = \frac{1}{p_1 + q_1}U\left(W + \frac{1}{2}(\alpha_1 + \beta_1 + \gamma_1 + \delta_1)\mathbb{I}\right)U^{-1}. \tag{B.5}$$

When dealing with two models to perform a mapping, we attach an integer $j = 1, 2$ to label the corresponding parameters, e.g. $p_j, q_j$ or $\rho_a^{(j)}, \sigma_b^{(j)}$, etc.

The constraints (4.8) and (4.9) then read

$$\sigma_a^{(1)} = \sigma_a^{(2)} \quad \text{and} \quad \sigma_b^{(1)} = \sigma_b^{(2)}, \tag{B.6}$$

meaning that the parameters $\sigma_a$ and $\sigma_b$ are conserved in the mapping. Thus, instead of having four boundary parameters $\sigma_{a,b}^{(j)}$, there are only two independent ones, $\sigma_a$ and $\sigma_b$ when considering the mapping.

This result can be exploited to reformulate the mapping between the scaled cumulant generating functions of two ASEPs using a minimal number of parameters, also taking into account the bulk driving force $\mathcal{E}_j = \log\sqrt{\frac{q_j}{p_j}}$ instead of the two bias parameters $p$ and $q$ separately. Equality (4.14) can then be rewritten as

$$\frac{1}{p_1 + q_1}\psi(\mu_1, s_1 | \mathcal{E}_1, \rho_a^{(1)}, \rho_b^{(1)}, \sigma_a, \sigma_b) + \frac{\sigma_a}{2\rho_a^{(1)} - 1} + \frac{\sigma_b}{2\rho_b^{(1)} - 1} + (\rho_a^{(1)} + \rho_b^{(1)} - 1)\tanh\mathcal{E}_1$$
$$= \frac{1}{p_2 + q_2}\psi\big(\mu_1 - E_1 + E_2\,,\, s_1 - \log\cosh(\mathcal{E}_1) + \log\cosh(\mathcal{E}_2)\big|\mathcal{E}_2, \rho_a^{(2)}, \rho_b^{(2)}, \sigma_a, \sigma_b\big)$$
$$+ \frac{\sigma_a}{2\rho_a^{(2)} - 1} + \frac{\sigma_b}{2\rho_b^{(2)} - 1} + (\rho_a^{(2)} + \rho_b^{(2)} - 1)\tanh\mathcal{E}_2. \tag{B.7}$$

Then, the remaining constraints (4.10) allow us to determine $\sigma_a$ and $\sigma_b$ as functions of the reservoir densities and the bulk driving forces $\mathcal{E}_j = \log\sqrt{\frac{q_j}{p_j}}$, $j = 1, 2$:

$$\sigma_a = \frac{1}{4}\frac{(\chi_a^{(1)} - 1)\sinh^2(\mathcal{E}_1) - (\chi_a^{(2)} - 1)\sinh^2(\mathcal{E}_2)}{\big(1 - (\chi_a^{(1)})^{-1}\big)\cosh^2(\mathcal{E}_1) - \big(1 - (\chi_a^{(2)})^{-1}\big)\cosh^2(\mathcal{E}_2)}, \tag{B.8}$$

$$\sigma_b = \frac{1}{4}\frac{(\chi_a^{(1)} - 1)\sinh^2(\mathcal{E}_1) - (\chi_a^{(2)} - 1)\sinh^2(\mathcal{E}_2)}{\big(1 - (\chi_b^{(1)})^{-1}\big)\cosh^2(\mathcal{E}_1) - \big(1 - (\chi_b^{(2)})^{-1}\big)\cosh^2(\mathcal{E}_2)}, \tag{B.9}$$

where we have introduced $\chi^{(j)} = (2\rho^{(j)} - 1)^2$, for $j = 1, 2$.

Although this reduction of parameters may be found elegant in some way, its drawback is that the expressions (B.8) and (B.9) of $\sigma_a$ and $\sigma_b$ respectively, involve parameters of both the source and target models of the mapping, which may not be very convenient in practice.

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
