# Peer review of "Mapping current and activity fluctuations in exclusion processes: consequences and open questions"

_SciPost Physics, doi:SciPost Phys. 10, 028 (2021)_

## Round 1 · Referee Report · Anonymous (Referee 1) · 2021-1-3

Strengths

1) Important new mapping for the ASEP

2) New formulas for LDF of activity

3) Relation with MFT and Spin Chains

4) A difficult open problem is raised in the diffusive regime. Can be viewed as a challenge in the field.

Report

This manuscript is a very good piece of work. The authors have discovered a very simple mapping that relates two deformed generators of the ASEP with different parameters (asymmetries, fugacities of activity and current), both in the periodic and the (finite) open case. As far as I can tell, this elementary observation was unknown (which is almost embarrassing!): it allows the authors to draw many new relations and a lot of relevant information about exclusion processes. Many known results, scattered in the literature of the last dozen years, can be restated within this new perspective and appear now in a coherent framework. In particular, I have found the section 3-7 (that relates the structure factor in the hyperuniform phase with ground state spin-spin correlations) particularly appealing. Finally, this work, besides its original results, also raises and states precise and important questions, that set interesting challenges for the specialists.

\vskip 0.3cm

Here are some minor remarks on the manuscript:

\vskip 0.1cm

1) Section 3-1 describes various scaling regimes. In could be useful if the authors could draw a table to summarize these results. \vskip 0.1cm

\vskip 0.1cm 2) The expansion of ${\mathcal K}(s)$ mentioned after (3.20) is not clear.

\vskip 0.1cm 3) Can the KPZ scaling mentioned after (3.23) be related to a property of the spectrum of SSEP ?

\vskip 0.1cm 4) What do the authors mean when they say after eq. (3-30) that they checked numerically the existence of the divergence by diagonalizing the evolution operator.

\vskip 0.1cm 5) A reference for eqs (4.18) could be useful.

\vskip 0.1cm \vskip 0.1cm I recommend publication of this manuscript in SCiPost without hesitation.

---

## Round 1 · Referee Report · Anonymous (Referee 2) · 2021-1-20

Strengths

  1. Expands and completes our understanding of dynamical large deviations in simple exclusion processes (a fundamental basic model of non-equilibrium), specifically for fluctuations of the dynamical activity and current in the large deviation regime.
  2. It does so via a simple (and very useful) relation between the tilted generators (for activity and current) of different SEPs, both for the periodic boundaries and for boundary driven cases.
  3. It aggregates a decade or more of work in this problem. The paper is both conceptually clear and of high technical standard, combining exact methods for spin chains with MFT in the weakly asymmetric SEP.
  4. Reveals a KPZ regime for activity fluctuations in the symmetric SEP.
  5. Relates to the growing list of mappings in trajectory ensembles of non-equilibrium systems, such as Doob and gauge transformations, or mappings between equilibrium and non-equilibrium.

Report

This paper deals with an important problem in non-equilibrium statistical mechanics, the precise characterization of fluctuations in the dynamics. The natural framework for this question is the method of large deviations, and simple exclusion processes (SEPs) are fundamental models. Perhaps "completes" is too strong a word as some questions remain, but the paper goes a long way to fully complement and clarify over a decade of work on the LDs in SEPs, work that has ranged from exact methods for spin chains, in the diffusive regime via MFT, and numerical simulations. How this fits all together is given in the summary of known and new results in Sec.3.1. This is an excellent piece of work of high standard. I recommend publication as is.

---

## Round 2 · Author Response

Reply to Referee 1:

We are grateful to the referee for their positive assessment of our manuscript.

Referee’s comment: 1) Section 3-1 describes various scaling regimes. In could be useful if the authors could draw a table to summarize these results.

Our answer: We have improved the presentation of the list so that it clearly distinguishes the different regimes in s. We believe it will play the role of the table suggested by the referee, by gathering the definitions of the regimes of $s$ and the behaviour of the SCFG in a summarized manner.

Referee’s comment: 2) The expansion of $\mathcal K(s)$ mentioned after (3.20) is not clear.

Our answer: We meant that the function $\mathcal K(s)$ can be expanded as a power series of $\sqrt{-s}$, and we agree that the justification for such an expansion was not clearly stated. We have reformulated the two paragraphs after (3.21) in order to make this issue clearer.

Referee’s comment: 3) Can the KPZ scaling mentioned after (3.23) be related to a property of the spectrum of SSEP?

Our answer: we thank the referee for this interesting remark; it is true that the gap between the maximal eigenvalue of the operator and the next one has been computed in varied systems (the TASEP and the ASEP), and is found to scale as $1/L^{3/2}$ (refs [73,126,127] of the resubmitted version), when there are no Legendre parameters conjugated to the current or the activity. Extending their analysis of the gap to our problem in the presence of a bias is beyond the scope of our work, but we have added a remark in the conclusion where we had already mentioned the gap. We stress in particular that a $1/L^{3/2}$ gap in the deformed matrix M of the activity in the SSEP would imply a $3/2$ dynamical exponent for large enough $|s|$ in this model.

Referee’s comment: 4) What do the authors mean when they say after eq. (3-30) that they checked numerically the existence of the divergence by diagonalizing the evolution operator?

Our answer: We have considered systems with values of $N$ and $L$ up to $16$. To compute $D$, we used the relation $D=\lim_{t\to\infty} \frac{\langle Q^2\rangle_c}{t}$ mentioned before (3.29). This expression can be evaluated numerically from the second derivative of the SCGF, which is the maximal eigenvalue of the matrix $W$. This eigenvalue was evaluated by numerically diagonalizing $W$ with arbitrary numerical precision (allowing to take the second derivative numerically). When varying $\mu'$ from $0$, the numerically obtained value of $D$ diverges when $\mu'\to\mu'_c$. We have updated the paragraph after (3.30) in order to explain this procedure.

Referee’s comment: 5) A reference for eqs (4.18) could be useful.

Our answer: References [113,114] have been added.

Reply to Referee 2:

We thank the referee for their careful reading of the manuscript and very positive appraisal of our work. We are glad that the referee "recommend[s] publication as is".

---

## Round 2 · List of Changes

§3.1:
. In the summary on page 8, titles of items are highlighted in a better way.
. In II), the definition of "finite $\lambda$" has been made explicit.

§3.3: after (3.21), the series expansion of $\mathcal K(s)$ in powers of $\sqrt{-s}$ has been justified in the paragraph before (3.22).

§3.5: the details of the numerical procedure to evaluate D and its divergence as $\mu'\to\mu'_c$ have been made explicit

§4.2: before (4.18), references [113,114] have been added

§5: in the last paragraph of the conclusion, the discussion on the gap has been expanded, conjecturing a possible KPZ-type dynamical exponent in the SCGF of the SSEP for large-enough deviations of the activity.

---

## Editorial Decision

published